# GLARE: Scalable Neuro-Symbolic Reward Shaping for LLM Agents via Group-Level Automata

**Jingyuan Yan** [1]   **Qingchen Liu** [1]   **Qichao Ma** [1]   **Jiahu Qin** [1]

## Abstract

Reinforcement Learning (RL) with Group Relative Policy Optimization (GRPO) shows great promise for enhancing LLM reasoning, but remains challenged by sparse and unstable rewards in long-horizon tasks. Existing approaches to reward shaping struggle to balance semantic expressiveness, reliability, and computational efficiency: heuristic rules lack flexibility, while LLM-as-a-Judge incurs high computational cost and suffer from inconsistent and misaligned scoring signals in long-context settings. To address these challenges, we introduce GLARE, a neuro-symbolic reward framework that decouples semantic abstraction from credit assignment. Specifically, to leverage semantic understanding while preserving symbolic determinism, we first extract and symbolize trajectory events into a discrete representation. These events are then translated into Linear Temporal Logic (LTL) formulas, which are compiled into deterministic automata that track the agent's progress via state transitions. This mechanism yields dense and consistent reward signals, avoiding unstable direct scoring while significantly reducing computational cost. Empirical results on ALFWorld show that GLARE outperforms GRPO by 12.1% in success rate, while achieving an 8.1% improvement over conventional LLM-based judges using only 15% of their computational cost.

## 1. Introduction

Large Language Models (LLMs) (Achiam et al., 2023; Yang et al., 2025; Guo et al., 2025) have evolved from static knowledge engines into versatile agents capable of perceiving, reasoning, and acting within dynamic, open-ended en-

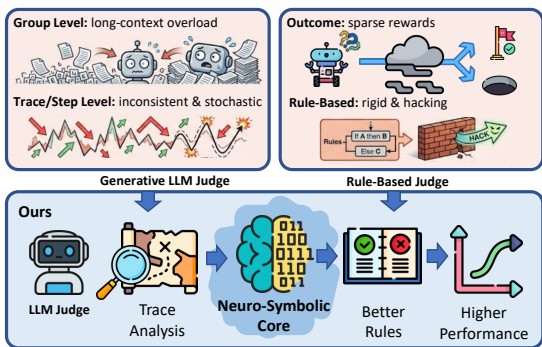

*Figure 1.* Existing reward schemes suffer from different shortcomings, including sparse outcome-level rewards, rigid and hackable rule-based judges, and inconsistent or unstable evaluations from LLM-based judges. We propose a neuro-symbolic reward paradigm that combines the semantic understanding of LLMs with the determinism of temporal logic, leveraging the strengths of both.

vironments. From abstract tasks including code (Jimenez et al., 2023) and math (Hendrycks et al., 2021), to navigating in simulated worlds (Shridhar et al., 2021; Yao et al., 2022), these agents are required to engage in multi-turn decision-making loops. Unlike single-turn tasks, these long-horizon scenarios demand not only linguistic comprehension but also the ability to adjust strategies based on environmental feedback over extended trajectories.

Reinforcement Learning (RL) has emerged as the standard paradigm for post-training LLMs, culminating in reasoning-heavy models like OpenAI o1 (OpenAI, 2024) and DeepSeek R1 (Guo et al., 2025). Group-based RL algorithms, such as GRPO (Shao et al., 2024) and DAPO (Yu et al., 2025), have demonstrated remarkable efficiency by estimating relative advantages without costly value networks. However, their success has been largely confined to domains like mathematics and coding (Yang et al., 2025; Shao et al., 2024), where ground-truth verifiers (e.g., compilers or symbolic solvers) provide precise, deterministic feedback. In contrast, agentic tasks in external environments suffer from the sparse reward problem, where credit assignment becomes intractable without dense, intermediate signals.

To tackle credit assignment, prior works have relied on rule-based heuristics (Qian et al., 2025) or process veri-

[1]University of Science and Technology of China, Hefei, Anhui, China. Correspondence to: Jiahu Qin <jhqin@ustc.edu.cn>.

*Proceedings of the 43rd International Conference on Machine Learning*, Seoul, South Korea. PMLR 306, 2026. Copyright 2026 by the author(s).

fiers (Yang et al., 2025), yet these approaches often lack the semantic adaptability required for open-ended agents. The "LLM-as-a-Judge" paradigm (Gu et al., 2024) addresses this gap but faces a critical dilemma in long-horizon RL. The inherent limitations of LLMs are further amplified when they are employed as judges. Prior work has shown that LLM-based evaluators suffer from position bias, evaluation inconsistency (Ye et al., 2025), and discrepancies between intermediate reasoning and final judgments (Turpin et al., 2023), as well as attribute-binding errors, leading to unreliable scoring signals. Crucially, these biases introduce inconsistent evaluation standards within comparison groups, which undermines the estimation of relative advantages that is essential for policy optimization (Coste et al., 2023). As a result, the induced noise in reward signals can interfere with reinforcement learning training.

To leverage the semantic understanding of LLMs while preserving the determinism of symbolic rules, we decouple trajectory analysis requiring semantic understanding from step-wise verification demanding deterministic logic, and introduce GLARE, a neuro-symbolic reward modeling framework. Specifically, given a group of agent trajectories, we first extract and symbolize a sequence of low-level events, yielding a discrete symbolic representation. From this representation, a lightweight semantic abstraction module identifies a set of salient semantic predicates that characterize task-relevant phenomena along the trajectory. These predicates are then composed into Linear Temporal Logic (LTL) formulas encoding the desired task constraints. Each LTL specification can be compiled into a deterministic automaton, which tracks an agent's progress toward satisfying the task and provides reward signals based on its state transitions. This construction yields a dense and well-defined reward signal that depends only on automaton transitions, thereby avoiding unstable direct semantic scoring and ensuring consistency across different trajectories with significantly reduced computational complexity.

We evaluate our method on a challenging long-horizon benchmark: ALFWorld (Shridhar et al., 2021), which involves multi-step reasoning and strong temporal dependencies. Results demonstrate that GLARE significantly outperforms GRPO (+12.1%), and achieves better performance (+8.1%) than conventional LLM-based reward modeling baselines while using only 15% of their computational cost.

## 2. Related Works

### 2.1. Agentic Reinforcement Learning

Recent work has explored training large language models (LLMs) as agents that interact with environments through predefined action or tool spaces (Zhang et al., 2025). These agents receive rich environmental feedback as observations

and have demonstrated promising results in Code (Yang et al., 2025), search (Jin et al., 2025), and general-purpose device control (Shi et al., 2025). When dealing with long-horizon interaction tasks such as ALFWorld (Shridhar et al., 2021), they naturally present challenges for reinforcement learning in credit assignment. To address this, some approaches rely on manually designed reward rules (Qian et al., 2025), while others approach leverages unsupervised signals derived from model uncertainty or self-consistency(Agarwal et al., 2025; Zuo et al., 2025), frequently quantified by policy entropy, though a consensus is still lacking. Another line of work follows the process reward models paradigm (Lightman et al., 2023). However, such reward models are often trained for specific domains and require substantial training data, making them expensive to train (Zhang et al., 2024; Xi et al., 2025). Alternatively, employing LLM-based evaluators to provide process-level supervision is often expensive and suffers from the inherent biases associated with "LLM-as-a-Judge" frameworks (Ye et al., 2025).

### 2.2. Temporal Logic and Automata in Policy Learning

Linear Temporal Logic (LTL) and automata theory provide a rigorous framework for specifying complex, non-Markovian task constraints (Baier & Katoen, 2008). In the context of classical reinforcement learning and robotics, compiling LTL specifications into Finite State Automata (FSA) is a well-established technique to guide exploration (Icarte et al., 2018), ensure safety (Alshiekh et al., 2018), and provide progress-based or hybrid reward shaping (Li et al., 2017; Kwon et al., 2025). Recent work also explores using foundation models to construct reward machines from task specifications (Castanyer et al., 2026).However, the integration of these formalisms with LLM agents remains limited. Existing works typically focus on either translating natural language to LTL for instruction following (Pan et al., 2023; Chen et al., 2023) or applying static constraints within fixed environments (Wang et al., 2024; Khan et al., 2025). Neither approach effectively leverages logic to guide optimization in open-ended semantic environments.

## 3. Preliminaries

In this section, we formally define the problem of long-horizon agentic planning, review the group relative policy optimization, and introduce the necessary background on Linear Temporal Logic (LTL) and automata theory.

### 3.1. Problem Setup

We formulate the agentic task as a Partially Observable Markov Decision Process (POMDP), defined by the tuple $\mathcal{M} = \langle \mathcal{S}, \mathcal{A}, \mathcal{O}, \mathcal{T}, \mathcal{R}, \gamma \rangle$, **where** $\mathcal{S}$ denotes the hidden state space, and $\mathcal{A}$ represents the action space consisting of natural language strings. At each time step $t$, the agent receives

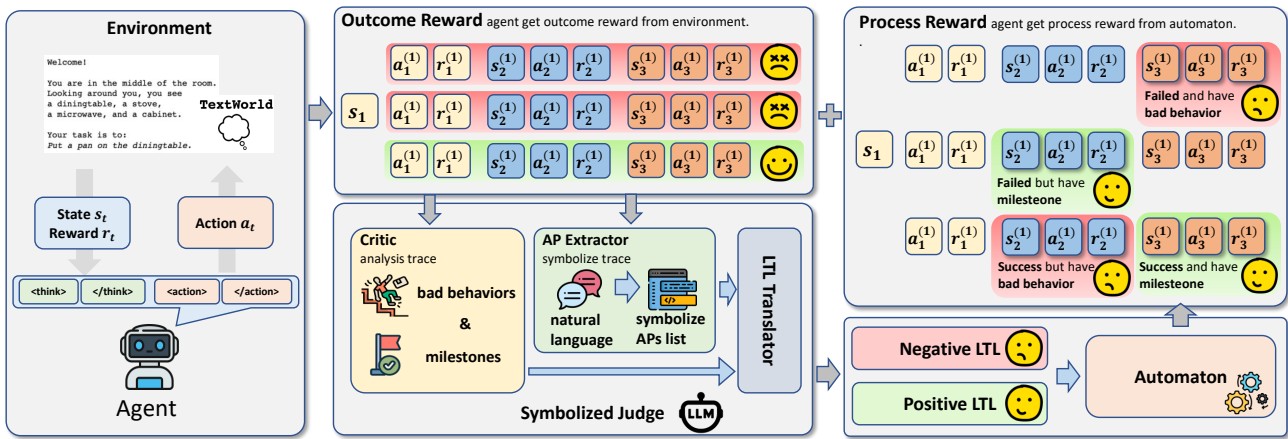

*Figure 2.* Overview of the GLARE framework. Left: An agent interacts with the environment, producing a trajectory of state–action pairs $(s_t, a_t)$. Middle (top): A sparse outcome-level reward is assigned based on task success. Middle (bottom): The core analysis module of GLARE performs semantic analysis over trajectories, extracts atomic propositions, and synthesizes Linear Temporal Logic (LTL) formulas. Right: The reward annotation module of GLARE compiles the synthesized LTL formulas into deterministic automata, which re-scan the symbolic trajectories to assign dense process rewards.

an observation $o_t \in \mathcal{O}$ derived from the underlying state $s_t \in \mathcal{S}$, which corresponds to the textual feedback from the environment.

Since the full state $s_t$ is inaccessible, the agent relies on the interaction history $h_t = (o_0, a_0, \ldots, a_{t-1}, o_t)$ to make decisions. The policy $\pi_\theta(a_t|h_t)$, parameterized by an LLM $\theta$, maps the current history to a distribution over actions. The environment provides a sparse extrinsic reward $R_{env}(s_t, a_t)$, typically non-zero only upon successful task completion. The objective of the agent is to maximize the expected cumulative return:

$$J(\theta) = \mathbb{E}\tau \sim \pi\theta \left[ \sum_{t=0}^{T} \gamma^t R_{env}(s_t, a_t) \right], \qquad (1)$$

where $\tau = (s_0, o_0, a_0, \ldots)$ represents a trajectory and $\gamma \in [0, 1]$ is the discount factor. In long-horizon tasks like ALFWorld, $T$ can be large, and $R_{env}$ is extremely sparse, making optimization via standard Reinforcement Learning (RL) intractable without dense shaping signals.

### 3.2. Group Relative Policy Optimization (GRPO)

We employ Group Relative Policy Optimization (GRPO) (Shao et al., 2024) as our core learning algorithm. Unlike standard Proximal Policy Optimization (PPO) (Schulman et al., 2017), which relies on a separate value network (Critic) to estimate the state-value function $V(s)$, GRPO is critic-free. It estimates the baseline directly from the group statistics of sampled trajectories, significantly reducing memory overhead and computational cost.

Formally, for each observation $x$, GRPO samples a group of $G$ outputs $\{y_1, \ldots, y_G\}$ from the reference policy $\pi_{\theta_{old}}$.

The optimization objective maximizes the following surrogate loss:

$$\mathcal{J}_{GRPO}(\theta) = \mathbb{E} \left[ \frac{1}{G} \sum_{i=1}^{G} \min \left( \rho_i A_i, \tilde{\rho}_i A_i \right) - \beta \mathbb{D}_{KL} \right] \quad (2)$$

where $\tilde{\rho}_i = \text{clip}(\rho_i, 1 - \epsilon, 1 + \epsilon)$ represents the clipped probability ratio preventing excessive policy updates. The expectation $\mathbb{E}$ is taken over task prompts $x \sim \mathcal{D}$ and the sampled group. Here, $\rho_i = \frac{\pi_\theta(y_i|x)}{\pi_{\theta_{old}}(y_i|x)}$ is the importance sampling ratio, and $\mathbb{D}_{KL}$ denotes the KL-divergence regularization term.

Crucially, the advantage $A_i$ for the $i$-th trajectory is computed by normalizing rewards within the group, utilizing the group mean as the baseline:

$$A_i = \frac{r_i - \text{mean}(\{r_1, \ldots, r_G\})}{\text{std}(\{r_1, \ldots, r_G\}) + \epsilon} \quad (3)$$

This formulation implies that optimization quality is entirely dependent on the accuracy of the scalar reward $r_i$. In deterministic domains like coding, $r_i$ is derived from ground-truth verifiers. However, in open-ended agentic tasks, prior works typically resort to the "LLM-as-a-Judge" paradigm (Gu et al., 2024). As discussed, these surrogate signals often suffer from high variance, numeracy limitations, and reasoning-answer inconsistency. Our method aims to replace these stochastic signals with deterministic, logic-guided verifiers to stabilize the GRPO training loop.

### 3.3. LTL Formula and Automaton

**Linear Temporal Logic (LTL).** LTL is a formalism used to describe properties of paths in a transition system. An LTL

formula $\varphi$ is defined over a set of Atomic Propositions $AP$. The syntax is given by:

$$\varphi ::= \top \mid p \mid \neg\varphi \mid \varphi_1 \wedge \varphi_2 \mid X\varphi \mid F\varphi \mid G\varphi \mid \varphi_1 U\varphi_2, \quad (4)$$

where $p \in AP$. The operators include standard Boolean connectives and temporal operators: $X$ (Next), $F$ (Eventually), $G$ (Globally), and $U$ (Until). For example, a milestone requirement "eventually open the door" can be expressed as $F(\text{door\_open})$, and a constraint "never repeat an action" as $G(\neg\text{repeat})$. The satisfaction of a formula $\varphi$ by a trace $\sigma = (\sigma_0, \sigma_1, \dots)$, denoted as $\sigma \models \varphi$, is determined by the sequence of truth assignments of $AP$ at each step.

**Finite State Automaton (FSA).** Any LTL formula $\varphi$ over $AP$ can be translated into a deterministic Finite State Automaton (FSA) (or a Büchi Automaton for infinite traces). We define the corresponding automaton as a tuple $\mathcal{A}_\varphi = \langle Q, \Sigma, \delta, q_0, \mathcal{F} \rangle$, where:

- $Q$ is a finite set of automaton states.

- $\Sigma = 2^{AP}$ is the alphabet consisting of subsets of atomic propositions.

- $\delta : Q \times \Sigma \to Q$ is the deterministic transition function.

- $q_0 \in Q$ is the initial state.

- $\mathcal{F} \subseteq Q$ is the set of accepting states.

In GLARE, we utilize the automaton to monitor the agent's progress. At each step $t$, the current automaton state $q_t$ transitions to $q_{t+1} = \delta(q_t, L(s_t))$ based on the extracted propositions $L(s_t) \subseteq AP$. This enables precise monitoring of the agent's progress, triggering rewards (or penalties) whenever specific semantic conditions defined by $\varphi$ are met.

## 4. Method

In this section, we present GLARE, a neuro-symbolic framework designed to provide dense, reliable, and consistent rewards for long-horizon agentic tasks. We model the problem as a Partially Observable Markov Decision Process (POMDP). The core pipeline consists of two components: (1) Symbolic Grounding via Explicit State Tracking, (2) Logic-Guided Reward Generation. Figure 2 presents an overview of the GLARE training pipeline.

### 4.1. Symbolic Grounding via Explicit State Tracking

To bridge the gap between unstructured natural language observations and formal logic verification, we propose a rigorous Symbolic Grounding mechanism. Unlike prior works (Li et al., 2024) that rely on large models to directly extract Atomic Propositions (APs)—an approach often constrained by inconsistent formal representations and

a heavy reliance on extensive, environment-specific customization—we introduce a Decomposed Triple Extraction strategy coupled with an Explicit Belief State Tracker.

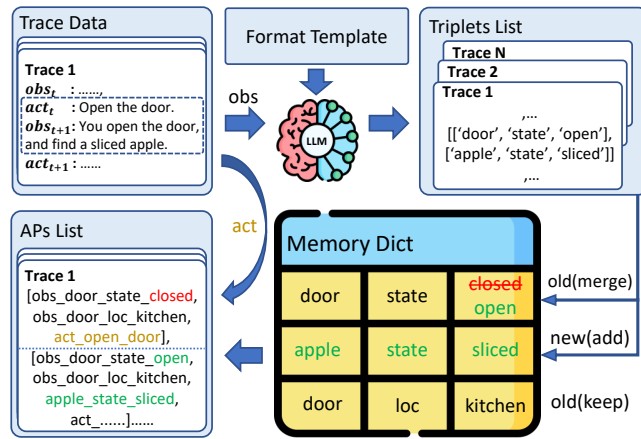

*Figure 3.* Example of extracting APs list from natural language trajectories via explicit state tracking

**Decomposed Extraction.** We separate the processing of agent actions and environmental observations. **1) Actions**: Since the agent's output is deterministic, we directly map actions $a_t$ to propositions using rule-based string matching (e.g., act\_open\_door1). **2) Observations**: For the open-ended observation $o_t$, we employ a lightweight LLM (e.g., 4B) to extract a set of structured triples. Let $\mathcal{K}, \mathcal{C}, \mathcal{V}$ be the sets of subjects, categories, and values. Crucially, to circumvent the manual design of rigid, environment-specific templates, we employ a few-shot prompting strategy. We provide generic demonstrations to guide the extraction logic—teaching the model how to parse entity-state relations—while permitting open-ended generation for the specific content of $k, c, v$. The LLM approximates an extraction function $E_\phi : \mathcal{O} \to 2^{\mathcal{K} \times \mathcal{C} \times \mathcal{V}}$, mapping $o_t$ to a set of triples $\{\tau_1, \dots, \tau_m\}$ where $\tau = (k, c, v)$. Furthermore, to enhance efficiency and consistency, we implement Group Batching, where identical observations across the group share the extraction computation.

**Explicit State Tracking.** To maximize computational efficiency, we execute the triplet extraction independently and in parallel across all time steps. To handle the "memoryless" nature of independent extraction and prevent state conflict, we maintain an explicit belief state $h_t$, formulated as a partial function $h_t : \mathcal{K} \times \mathcal{C} \to \mathcal{V}$. The state update follows a Markovian Overwrite Rule, ensuring that the most recent observation overrides outdated information while preserving uncontradicted history:

$$h_t(k, c) = \begin{cases} v & \text{if } \exists(k, c, v) \in E_\phi(o_t) \\ h_{t-1}(k, c) & \text{otherwise} \end{cases} \quad (5)$$

This explicit tracking allows us to construct the final set of

Atomic Propositions $AP_t$ from $h_t$ without feeding the entire history to the LLM, significantly reducing computational cost and hallucination rates. A complete flowchart with an illustrative example is shown in Figure 3.

## 4.2. Logic-Guided Critic and LTL Generation

We propose a Logic-Guided Critic that acts as a meta-policy, translating group-level trajectory patterns into deterministic Linear Temporal Logic (LTL) formulas.

**Adaptive Group Analysis.** For a sampled group of trajectories $\mathcal{G} = \{\tau^{(i)}\}_{i=1}^G$, the Critic $\pi_{critic}$ analyzes the traces to extract semantic structures. Specifically, we instruct the LLM to perform a dual-track analysis: **1) Milestone Dependency Mapping:** It identifies pivotal steps that substantively contribute to the final goal and explicitly models their temporal dependencies (e.g., $A$ must precede $B$). Critically, to handle cases where all trajectories within a group fail, we introduce a counterfactual reasoning mechanism. Leveraging its global perspective and access to richer information than the acting agent, the Critic is guided to analyze the task description $\mathcal{I}_{task}$ and identify potential partial milestones latent in the observed trajectories. **2) Bad Behaviors Categorization:** It detects sub-optimal actions by grounding them into predefined bad behaviors categories, generating both a category tag and a natural language description. The categories and their descriptions are specified in Appendix A.3.

This structured tagging of temporal relations and problem pre-classification serves as a robust intermediate representation, significantly streamlining the subsequent translation into formal LTL formulas.

## 4.3. Formula Translation and Automaton Construction

Leveraging dependency-based semantic analysis, we translate extracted task signals into executable temporal logic specifications. Specifically, we distinguish two complementary classes of objectives: **Positive LTL (Pos-LTL, state-driven):** Encodes ordered task milestones as nested temporal goals (e.g., $F(m_1 \wedge F(m_2))$). All Pos-LTL clauses are composed into a single conjunctive specification, capturing the intended progression structure of the task. **Negative LTL (Neg-LTL, action-driven):** Encodes undesirable behaviors as safety constraints (e.g., $G(\neg \text{repeat})$). Each Neg-LTL clause is treated independently to ensure modular violation detection.

We leverage the Spot library (Duret-Lutz et al., 2016) to compile these specifications into automata. The composed Pos-LTL formula is translated into a **Deterministic Büchi Automaton (DBA)**, denoted as $\mathcal{A}^+$, which tracks the progressive satisfaction of task milestones. Conversely, each Neg-LTL constraint is compiled into an independent safety monitor $\mathcal{A}_j^-$, which immediately flags invariant violations.

During execution, automaton state transitions are efficiently evaluated via Binary Decision Diagrams (BDDs). Once generated, all automata are broadcast to every trajectory within the current group $\mathcal{G}$, ensuring that all traces are evaluated under a unified logical specification during each update step.

## 4.4. Automaton-Based Reward Assignment

We now describe how automaton execution induces dense and structured reward signals. Let the Pos-LTL automaton be denoted as $\mathcal{A}^+ = \langle Q, \Sigma, \delta, q_0, F \rangle$, where $Q$ is the state space, $q_0$ is the initial state, and $F$ is the accepting set. At each time step $t$, grounded atomic propositions $AP_t$ induce a transition $q_{t+1} = \delta(q_t, AP_t)$.

We define a *progressive transition* as any automaton transition that advances the state toward acceptance. Let $\mathcal{P}(q_t) \subset Q$ denote the set of such progressive successor states. To further mitigate reward sparsity, we define the shaping interval as $\mathcal{T}_{\text{trend}} = \bigcup_k \{t \mid \tau_{k-1} < t < \tau_k\}$, where $\tau_k$ denotes the timestamp of the $k$-th milestone update in $\mathcal{A}^+$. The step-level logic reward is defined as:

$$r_t^{\text{LTL}} = \underbrace{\lambda_{\text{pos}} \cdot \mathbb{I}\left(q_{t+1} \in \mathcal{P}(q_t)\right)}_{\text{Milestone Reward}} + \underbrace{\lambda_{\text{trend}} \cdot \mathbb{I}\left(t \in \mathcal{T}_{\text{trend}}\right)}_{\text{Trend Shaping}}$$
$$+ \underbrace{\lambda_{\text{neg}} \cdot \mathbb{I}\left(\exists j : \mathcal{A}_j^- \text{ rejects at } t\right)}_{\text{Violation Penalty}}. \tag{6}$$

The overall step-level reward combines environment feedback, logic-guided supervision, and syntactic regularization:

$$R_t^{\text{total}} = r_t^{\text{env}} + \beta \cdot r_t^{\text{LTL}} + r_t^{\text{fmt}}. \tag{7}$$

## 4.5. Step-Level Reward Normalization

Here, *step-level* refers to the granularity of reward assignment: each environment interaction step, i.e., each action–observation transition, receives a scalar reward. This is different from token-level or chain-of-thought-level supervision. For a prompt $x$, let $\mathcal{G}_x = \{\tau^{(i)}\}_{i=1}^G$ be the rollout group, where trajectory $\tau^{(i)}$ has length $T_i$. We normalize rewards over all valid environment steps in the group, i.e., $\mathcal{I}_x = \{(i, t) \mid 1 \le i \le G, 1 \le t \le T_i\}$, rather than aligning trajectories by the same timestamp.

Given the step-level total reward $R_{\text{total}, t}^{(i)}$ from Eq. (7), we compute $\mu_x$ and $\sigma_x$ as the mean and standard deviation of $\{R_{\text{total}, u}^{(j)} \mid (j, u) \in \mathcal{I}_x\}$. The normalized step-level advantage is then

$$A_t^{(i)} = \frac{R_{\text{total}, t}^{(i)} - \mu_x}{\sigma_x + \epsilon}, \qquad (i, t) \in \mathcal{I}_x. \tag{8}$$

The policy is optimized with the GRPO objective in Eq. (2), aggregating these advantages over all valid environment steps.

*Table 1.* Performance in ALFWorld. The results are averaged over 3 random seeds. Following the setup in the GiGPO baseline, we report the average success rate (%) for each subtask as well as the overall result. GRPO ( with an LLM-derived reward ) employs a trace-level "LLM-as-a-Judge" approach, which represents the best-performing instantiation of direct LLM based evaluation.

| Type | Method | ALFWorld | | | | | | |
|---|---|---|---|---|---|---|---|---|
| | | Pick | Look | Clean | Heat | Cool | Pick2 | All |
| *Closed-Source Model* | | | | | | | | |
| Prompting | GPT-4o | 75.3 | 60.8 | 31.2 | 56.7 | 21.6 | 49.8 | 48.0 |
| Prompting | Gemini-2.5-Pro | 92.8 | 63.3 | 62.1 | 69.0 | 26.6 | 58.7 | 60.3 |
| *Qwen2.5-1.5B-Instruct* | | | | | | | | |
| Prompting | Qwen2.5 | 5.9 | 5.5 | 3.3 | 9.7 | 4.2 | 0.0 | 4.1 |
| Prompting | ReAct | 17.4 | 20.5 | 15.7 | 6.2 | 7.7 | 2.0 | 12.8 |
| Prompting | Reflexion | 35.3 | 22.2 | 21.7 | 13.6 | 19.4 | 3.7 | 21.8 |
| RL Training | PPO (with critic) | $64.8_{\pm3.5}$ | $40.5_{\pm6.9}$ | $57.1_{\pm4.9}$ | $60.6_{\pm6.6}$ | $46.4_{\pm4.0}$ | $47.4_{\pm1.9}$ | $54.4_{\pm3.1}$ |
| RL Training | RLOO | $88.3_{\pm3.0}$ | $52.8_{\pm8.6}$ | $71.0_{\pm5.9}$ | $62.8_{\pm8.7}$ | $66.4_{\pm5.5}$ | $56.9_{\pm4.7}$ | $69.7_{\pm2.5}$ |
| RL Training | GRPO | $85.3_{\pm3.0}$ | $64.1_{\pm2.6}$ | $80.3_{\pm7.0}$ | $84.4_{\pm8.0}$ | $77.8_{\pm1.6}$ | $41.7_{\pm9.3}$ | $72.4_{\pm1.0}$ |
| LLM Reward | **GRPO** | $91.4_{\pm2.8}$ | $54.5_{\pm5.2}$ | $75.0_{\pm9.4}$ | $84.3_{\pm2.0}$ | $64.4_{\pm8.3}$ | $77.8_{\pm3.7}$ | $76.4_{\pm2.9}$ |
| LLM Reward | **GLARE** | $94.7_{\pm2.8}$ | $70.5_{\pm3.5}$ | $87.9_{\pm2.5}$ | $88.2_{\pm5.9}$ | $82.4_{\pm7.5}$ | $73.6_{\pm6.1}$ | $\mathbf{84.5}_{\pm2.9}$ |
| *Qwen2.5-7B-Instruct* | | | | | | | | |
| Prompting | Qwen2.5 | 33.4 | 21.6 | 19.3 | 6.9 | 2.8 | 3.2 | 14.8 |
| Prompting | ReAct | 48.5 | 35.4 | 34.3 | 13.2 | 18.2 | 17.6 | 31.2 |
| Prompting | Reflexion | 62.0 | 41.6 | 44.9 | 30.9 | 36.3 | 23.8 | 42.7 |
| RL Training | PPO (with critic) | $92.3_{\pm4.0}$ | $64.0_{\pm8.4}$ | $92.5_{\pm2.4}$ | $89.5_{\pm7.0}$ | $80.3_{\pm2.0}$ | $68.8_{\pm8.3}$ | $80.4_{\pm2.7}$ |
| RL Training | RLOO | $87.6_{\pm4.3}$ | $78.2_{\pm8.3}$ | $87.3_{\pm5.8}$ | $81.3_{\pm7.6}$ | $71.9_{\pm5.2}$ | $48.9_{\pm8.4}$ | $75.5_{\pm4.6}$ |
| RL Training | GRPO | $88.7_{\pm1.6}$ | $54.5_{\pm1.0}$ | $80.4_{\pm1.8}$ | $82.4_{\pm5.9}$ | $72.4_{\pm10.3}$ | $62.5_{\pm6.5}$ | $75.7_{\pm0.7}$ |
| LLM Reward | **GRPO** | $94.2_{\pm5.8}$ | $80.1_{\pm10.8}$ | $80.6_{\pm6.9}$ | $87.1_{\pm0.4}$ | $80.7_{\pm7.0}$ | $64.2_{\pm5.8}$ | $82.4_{\pm1.2}$ |
| LLM Reward | **GLARE** | $97.6_{\pm0.0}$ | $80.8_{\pm11.5}$ | $89.9_{\pm4.9}$ | $90.0_{\pm3.3}$ | $83.3_{\pm2.4}$ | $72.5_{\pm2.5}$ | $\mathbf{86.7}_{\pm0.8}$ |

## 5. Experiment

This section describes the benchmarks and baselines, including conventional RL and LLM-based reward modeling methods, followed by experimental results. Hyperparameters and additional settings are provided in Appendix A.2.

### 5.1. Benchmarks & Model

We train the LLM agents on a challenging benchmarks: ALFWorld (Shridhar et al., 2021). *ALFWorld* is an embodied environment designed to assess the ability of LLM agents to perform multi-step decision-making. In each episode, the agent receives a text goal and must accomplish it through multi-turn interaction with the environment. It includes 3,827 task instances across six categories of common household activities: Pick & Place (Pick), Examine in Light (Look), Clean & Place (Clean), Heat & Place (Heat), Cool & Place (Cool), and Pick Two & Place (Pick2). We evaluate Qwen2.5-1.5B/7B-Instruct (Yang et al., 2024) on verl-agent (Feng et al., 2025) agent framework.

### 5.2. Baseline

To provide a comprehensive comparison, we adopt the baseline results reported in GiGPO and re-execute GRPO under our hardware settings for fair comparison.

To contrast standard neural evaluation with our neuro-symbolic approach, we implement two LLM-based judge baselines. To ensure a fair comparison, these baselines adopt the identical judging policy and prompting strategy as our method. The two baselines differ from each other only in the scope of input context: **1) Group-Level Direct Judge** receives the full sampled trajectory group as input and assigns step-wise binary reward/penalty markers to each trace based on the global context. **2) Trace-Level Direct Judge** applies the same judging procedure but evaluates each trajectory independently using only its own interaction history. We exclude the step-level judge due to its prohibitive computational cost ( 20 mins per step versus 100–200 seconds for other settings). Both LLM-based judge baselines further incorporate intermediate trending rewards between milestones and operate under an identical reward scale to ensure fair comparison.

### 5.3. Performance Result

The results in Table 1 demonstrate that incorporating mixed dense rewards—even when derived from a smaller external LLM—substantially enhances policy performance. On the 1.5B scale, the GRPO with Trace-level Judge[1] and GLARE-

---

[1]We early-stopped the group-level judge baseline as its performance was significantly inferior to other baseline and choose trace-

achieve relative gains of 4% and 12.1%, respectively; On the 7B scale, these margins expand to 6.7% and 11%, respectively. Crucially, GLARE consistently outperforms the Trace-level Judge baseline by 8.1% (1.5B) and 4.3% (7B), empirically underscoring the superiority of our neuro-symbolic framework over direct stochastic scoring.

A critical observation is the performance collapse in the Trace-level judge baseline. We observed that continued training after initial convergence leads to significant degradation, a phenomenon particularly pronounced in the 7B model. To ensure a rigorous baseline, we report its peak performance (recorded at step 100, prior to the onset of collapse) for the 7B setting. In contrast, GLARE exhibits no such instability, maintaining robust performance throughout training. We attribute this stability to the strict intra-group reward consistency guaranteed by our deterministic automata (detailed analysis in Sec. 6.1).

### 5.4. Tokens Usage Result

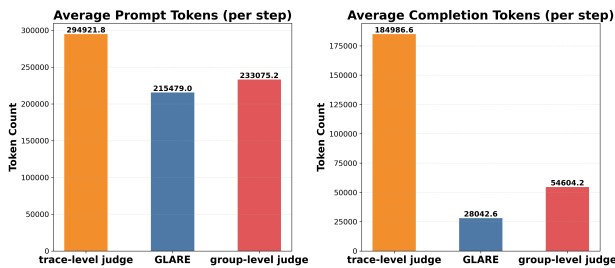

*Figure 4.* Computational cost analysis: Comparison of token usage between GLARE and direct LLM-based judges.

In addition to the significant improvements in task performance, our strategy of delegating reward assessment to automata offers a distinct advantage in computational efficiency. Figure 4 illustrates the token consumption compared to LLM-based judge baselines, measured over the first 50 training steps to normalize cost across methods with varying convergence speeds, as the policy's performance level directly influences the length of sampled trajectories. We observe a significant reduction in completion token usage compared to trace-level scoring (only 15% usage). Crucially, while GLARE significantly outperforms group-level baselines in success rate, it also surpasses trace-level baselines at a fraction of the computational cost. These results demonstrate that our neuro-symbolic paradigm is not only reliable but also highly efficient.

## 6. Analysis

In this section, we analyze the limitations of the LLM-as-a-Judge paradigm in group-based algorithm and examine

level judge to represent the prevailing LLM-as-a-Judge paradigm..

how our method addresses these challenges. We focus on the consistency of reward proxies within groups and the alignment between intermediate reasoning and final reward assignments. We further provide a quantitative stability audit of the sub-modules and validate component contributions through ablation studies.

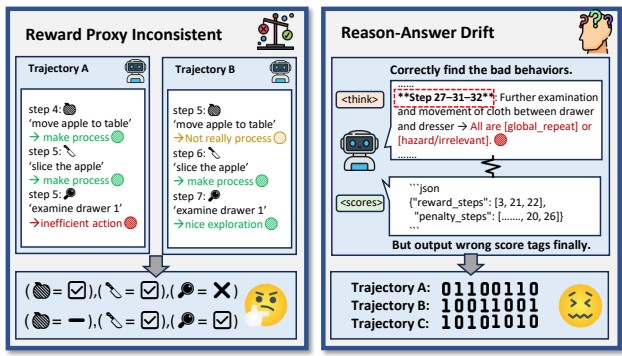

*Figure 5.* Visualizing the intrinsic failures of LLM-based judges. (Left) Reward Inconsistency: Identical or semantically similar behaviors receive contradictory evaluations across trajectories, leading to intra-group rank reversal. (Right) Reasoning-Answer Drift: A "Reasoning-Execution Gap" where the judge correctly identifies bad behaviors (Steps 27-32) in the reasoning trace but fails to ground these indices in the final structured output.

### 6.1. Intra-Group Consistency as a Requirement for Relative Policy Optimization

A common assumption is that denser, step-level rewards necessarily lead to better optimization. However, in group-based relative policy optimization, the stability of advantage estimation depends not only on reward granularity, but also critically on intra-group consistency. We provide a theoretical proof of this relationship in Appendix E.

Trace-level LLM judges provide dense supervision by evaluating individual steps, but each step is assessed independently through stochastic generation. As a result, the implicit evaluation standard varies across trajectories within the same group, introducing noise into relative advantage estimates. Group-level LLM judges, while enforcing a shared context, suffer from long-context degradation and attribute-binding errors, leading to inconsistent temporal grounding.

In contrast, our method enforces a shared reward logic by broadcasting a single LTL formula across all trajectories within a group. Although the induced rewards may be sparse or incomplete, the evaluation standard remains fixed, which is sufficient to preserve reliable relative ordering.

### 6.2. Reasoning-Answer Consistency

We further analyze the reliability of neural reward models by examining the alignment between their intermediate reasoning traces and final structured outputs. As shown

in Figure 5, both trace-level and group-level LLM judges exhibit reasoning–answer inconsistency when required to produce temporally grounded reward signals. Specifically, although the model often correctly identifies behavioral errors in its reasoning phase, it fails to consistently bind these errors to the correct time steps in the final output, resulting in off-by-one errors or hallucinated indices. This issue is exacerbated in group-level judges, where long-context processing and attribute binding further increase uncertainty.

Our neuro-symbolic framework avoids this failure mode by removing temporal grounding from neural generation altogether. Once semantic events are correctly identified, reward assignment is performed through deterministic execution of LTL formulas, ensuring consistent alignment between reasoning and reward signals.

### 6.3. Analysis for Sub-Module

In this section, we provide a quantitative analysis of the stability of the auxiliary components introduced in GLARE. We sampled 800 independent textual observations and 200 trajectory groups from the training logs, utilizing Gemini-3-Flash (Team et al., 2023) as the evaluator. Our evaluation targets two critical modules: 1) the **AP Extractor** for trajectory symbolization, and 2) the **LTL Translator** for converting semantic analysis into formal logic.

The AP extraction error rate is merely 4.4%, predominantly concentrated in specific symmetric relational patterns (e.g., confusing 'A loc B' with 'B loc A'). The information omission rate is low at 2.6%. Due to the independence of extraction at each step, we observed a slight intra-group redundancy (1.1%), where predicates are semantically identical but technically distinct in string format. During training, we employ regular expression–based formatting repairs together with a feedback-driven retry mechanism informed by compilation errors from the Spot library, and discard formulas that remain non-compilable. Consequently, our evaluation focuses strictly on semantic performance. We observed a 12% Mismatch Rate (invalid formulas) and only a 4% Harmful Rate. Error analysis suggests these failures stem primarily from the translator's instability in handling complex composite logic.

While most of these issues could be addressed through targeted supervised fine-tuning, our results demonstrate that the framework is already effective as a low-cost, training-free solution. Crucially, even in the presence of translation failures or information loss, GLARE faces reduced reward density but remains superior to GRPO. Achieving SOTA performance despite such sub-module imperfections highlights the intrinsic robustness of our framework. Detailed implementation specifications and an in-depth analysis of failure cases are provided in Appendix B. Further insights into the dynamic behavior of GLARE are discussed in Appendix D.

### 6.4. Ablation Study

Figure 6 presents an ablation study on Qwen2.5-1.5B-Instruct, dissecting the individual contributions of positive and negative reward components.

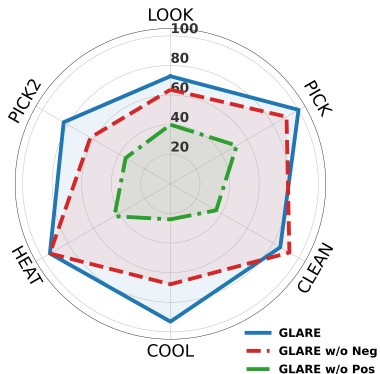

*Figure 6.* Ablation results. The y-axis shows success rate (%).

The Negative-Only variant underperforms even the baseline (which uses no external dense rewards). We attribute this to the lack of directional guidance: without a dominant positive signal to anchor the policy's optimization trajectory, purely punitive signals fail to shape effective behavior. Furthermore, given that our LTL-based penalty is not infallible, minor misjudgments—when unchecked by a robust positive signal—can induce severe training instability, causing the policy to degenerate. Conversely, the Positive-Only variant exhibits only a marginal performance degradation compared to the full method. However, this gap becomes significant in long-horizon tasks (e.g., Pick Two, Cool). This suggests that while positive milestones provide necessary high-level direction, they lack the granularity to prune suboptimal behaviors between milestones. The inclusion of smaller-scale negative rewards fills this gap by suppressing undesirable actions without disrupting the overall policy directionality. These findings confirm that the synergistic combination of dual reward signals—where positive rewards provide direction and negative rewards ensure safety—is essential for achieving optimal performance.

### 6.5. Generalization

To examine whether GLARE is tied to ALFWorld-specific observation patterns, we further evaluate it on WebShop, a web-browsing environment with different observation structures and task semantics. We keep the core Critic and LTL Translator unchanged. The only adaptation is in the AP Extractor: we replace the few-shot examples and relax the attribute category constraint to allow free-form product attributes such as color, size, and brand.

As shown in Table 2(a), GLARE improves over both GRPO and the LLM-judge baseline, suggesting that the proposed

*Table 2.* (a) WebShop tests cross-environment generalization using score and success rate. (b) The GiGPO comparison reports ALF-World success rates under original and perturbed observations.

| (a) WebShop Result | | | (b) Compare with GiGPO | | |
|---|---|---|---|---|---|
| Method | Score | Succ. | Setting | Method | Succ. |
| GRPO | 75.8 | 56.8 | Original | GiGPO | 86.7 |
| LLM Judge | 78.8 | 61.2 | Perturbed | GiGPO | 71.4 |
| GLARE | 80.2 | 63.4 | Perturbed | GLARE | 83.3 |

neuro-symbolic reward construction is not specific to ALF-World. Nevertheless, GLARE is best suited to object-centric agent tasks whose observations and actions can be reliably symbolized into event–object relations. When such symbolization is unavailable or meaningful event–object logic is difficult to define, extending GLARE may require task-specific symbolization or alternative event abstraction.

### 6.6. Relation to GiGPO

GiGPO is another strong method that improves LLM-agent training with fine-grained dense signals. In our additional comparisons, GiGPO outperforms GLARE on both ALF-World and WebShop under the original settings. However, its advantage relies heavily on state anchoring via exact observation matching. To highlight the complementary strength of GLARE, we further construct a perturbed ALF-World setting by injecting randomly sampled distractor descriptions into the original observations, while preserving the task-relevant object names and relations. This low-cost perturbation keeps the original task semantics unchanged but breaks the textual consistency required for exact state matching. As shown in Table 2(b), GiGPO loses its advantage under this perturbation, whereas GLARE remains largely unaffected. This suggests that GLARE provides a state-anchor-free reward construction that is robust to observation-level variation.

## 7. Conclusion

In this paper, we propose GLARE, a neuro-symbolic framework for credit assignment in long-horizon agentic tasks. By compiling group-level semantics into deterministic automata, GLARE enforces intra-group consistency and avoids reasoning–score misalignment from LLM evaluators. Experiments on ALFWorld show that GLARE outperforms both RL baselines and costly LLM judges, while achieving these gains at substantially lower computational cost, highlighting symbolic determinism as an efficient path for scalable agentic learning.

## Impact Statement

This paper presents work whose goal is to advance the field of Machine Learning. There are many potential societal consequences of our work, none which we feel must be specifically highlighted here.

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

# A. Implement Details

## A.1. Source Code

The implementation of GLARE is available at `https://github.com/USTC-AIR-Lab/GLARE.git`.

## A.2. Experiment Setting

**Base Training Configuration.** To ensure a fair comparison, we align our fundamental training setup with the GiGPO baseline. Specifically, we enforce a maximum prompt length of 2048 tokens and a response limit of 512 tokens, with each episode capped at 50 environment steps. Optimization is performed using a learning rate of $1e-6$ for the actor and $1e-5$ for the critic (PPO only). For group-based methods (GRPO, GLARE), we utilize a group size of $G = 8$ sampled across 16 rollouts, totaling 128 parallel environments, consistent with the 128 environments used in standard PPO. We maintain a rollout temperature of 1.0 and a validation temperature of 0.4. The mini-batch size is fixed at 256, with a KL-divergence coefficient of 0.01. Each experiment is trained for a total of 150 training iterations.

**Reward and Model-Specific Settings.** While we adopt the standard outcome reward of $+10$ for success and 0 for failure, we adjust the syntactic penalty to $-1$ to better align with our dense reward scale. For our proposed GLARE, the neuro-symbolic reward coefficients are set to $\lambda_{pos} = 2.0$ and $\lambda_{neg} = -0.5$ to serve as the primary behavioral drivers, supplemented by a smaller $\lambda_{trend} = 0.5$ for continuous shaping guidance. Regarding the auxiliary components (AP Extractor, Critic, and LTL Translator), we use Qwen3-4B-Instruct-2507-FP8 and set the sampling temperature to 1.0 to ensure diverse generation for robust extraction.

**Compute Infrastructure.** We tailor our hardware configuration to the model scale. For the **Qwen2.5-1.5B-Instruct** experiments, we utilize $2 \times$ NVIDIA H100 GPUs. To maximize computational resource utilization, we *co-locate* both the training policy and the reward model on the same set of GPUs. For the larger **Qwen2.5-7B-Instruct** experiments, we employ $4 \times$ NVIDIA A800 GPUs for policy training, with the reward model deployed on an additional, separate NVIDIA A800 GPU.

## A.3. Predefined Bad Behaviors Category

We classify bad behaviors into the following categories, designed to encompass the full range of error types present in general agent tasks. Below, we include explanations for each category along with their associated LTL translation templates:

- **sudden_repeat:** Immediately repeating the same action consecutively without sufficient justification. For example, performing action A and then immediately performing action A again is generally abnormal. **FORMAT:** `G(act_A -→ X(!act_A))`.

- **global_repeat:** Repeating an action or state sequence that offers no new information or progress, should be forbid forever. **FORMAT:** `G(act_A -→ G(!act_A))`.

- **lack_premise:** Executing an action without satisfying its prerequisite conditions. For example, action B only makes sense when item A is available, but the agent performs B prematurely. NOTICE: Exploratory behaviors do not have prerequisites and are not subject to this rule—be careful to distinguish them. **FORMAT:** `G(!Premise -→ !Subject)`.

- **hazard:** Irrational or potentially dangerous behaviors. **FORMAT:** `G(!act_Subject)`.

- **rollback:** Regressing task progress. For example, after acquiring a task-required item, discarding it before using it correctly—this is generally abnormal. **FORMAT:** `G(obs_Trigger -→ G(!act_Forbidden))`.

- **other:** Special issues that are difficult to classify into the above categories. **FORMAT:** `***`.

## B. Sub-Module Evaluation Details and Bad Case Analysis

In this section, we detail the quantitative evaluation methodology for the stability of the auxiliary sub-modules introduced in GLARE. Given that these tasks fall within the domain of Natural Language Processing (NLP), we still adopt LLM-as-a-Judge paradigm. Specifically, we design tailored prompts and employ a superior model (`Gemini-3-Flash`) to audit the reliability of the smaller models deployed in our framework. Following this, we present a qualitative analysis of error cases to provide intuitive examples of sub-module bottlenecks and discuss potential avenues for improvement.

### B.1. Atomic Proposition (AP) Extractor Evaluation:

To briefly recapitulate, Atomic Proposition (AP) extractor utilizes an LLM to extract relational triplets from open-ended semantic observations, serving as the foundation for subsequent AP synthesis. As the bedrock of environment symbolization, accurate triplet extraction is critical for the robust operation of GLARE.

To rigorously assess extraction quality, we evaluate the accuracy, coverage, and redundancy of the generated triplets. Specifically, we populate evaluation prompts (Figure 15 and Figure 16) with raw observations from training logs alongside their corresponding extracted triplets. The judge model evaluates performance across two ways: 1) Verifying the precision and completeness of triplet extraction within **individual observations** (i.e., ensuring no hallucinations and no missed details). We report the percentage of omitted triplets relative to the total as the **AP Missing Rate**, and the percentage of incorrect triplets as the **AP Error Rate**. 2) Detecting the presence of semantically identical but syntactically distinct predicates (i.e., non-strict string matches) within the aggregated AP list of a **group**. We report the proportion of such redundant predicates relative to the total AP count as the **AP Redundancy Rate**.

---

**Three Types of Failure Cases in AP Extractors**

```
[Missing Info Bad Case]
Observation : You close the microwave 1.
Triplets    : [["microwave 1", "state", "closed"]]
MISSING     : [["microwave 1", "seen", "true"]]
Reason      : The extraction correctly identified the state change, but missed the "
    seen" triplet which is standard for visible objects in this system.

[Error Extract Bad Case]
Observation : You arrive at drawer 2. The drawer 2 is open. In it, you see nothing.
Triplets    : [["self", "loc", "drawer 2"], ["drawer 2", "state", "open"], ["drawer
    2", "seen", "true"], ["drawer 2", "loc", "self"]]
BAD TRIPLETS: ["drawer 2", "loc", "self"]
Reason      : Triplet 3 ["drawer 2", "loc", "self"] is a hallucination/rule
    violation; the agent is not holding the drawer. The agent's location and the
    drawer's state are correctly identified.

[Redundancy Bad Case]
Redundant Sets:
[["obs_butterknife3_loc_countertop4", "obs_butterknive3_loc_countertop4"], ["
    obs_butterknife3_seen_true", "obs_butterknive3_seen_true"]]
Reason: The first two sets have spelling variations but refer to the same object and
     location. The last set describes the state of the cabinet, which cannot be both
    closed and open simultaneously.
```

*Figure 7.* Three types of failure cases found in the evaluation of AP extractors.

---

In addition to statistical metrics, we analyzed the primary patterns within failure cases and present representative examples across the three evaluation dimensions in Figure 7. Our detailed analysis yields the following insights:

1. **Triplet Extraction Issues:** Confusion predominantly arises with location-related predicates (`loc`), despite explicit emphasis in our prompts. Notably, omissions are almost exclusively related to the object property of "being seen" (e.g., *is_seen*). Critical, informative predicates are rarely missed. Since `seen`-type information is typically excluded from

LTL construction, we report the omission rate exclusively for non-`seen` predicates. We believe these minor issues can be effectively rectified through targeted supervised fine-tuning in future work.

2. **Intra-group AP Redundancy:** We observed that the model tends to generate divergent outputs when presented with varying noun forms. Although we mitigated this by merging identical observations, a small fraction of such inconsistencies persists. Future improvements could involve designing more robust extraction rules to ensure stricter consistency, or introducing a post-processing mechanism to consolidate redundant APs and re-map them to their corresponding trajectories.

### B.2. LTL Translator Evaluation:

LTL Translator is tasked with translating the structured outputs from the semantic analysis LLM into executable LTL specifications. We focus here on a semantic-level evaluation paradigm, applying a unified assessment logic to both Positive and Negative LTL generation. Specifically, we employ a superior LLM to cross-reference the original semantic analysis with the generated LTL formulas, verifying the translational fidelity of each entry. In instances of mistranslation, the judge further categorizes the error severity: distinguishing between Harmful Errors (which actively disrupt system operation) and Benign/Invalid Errors (which yield non-functional formulas with no adverse impact).

Accordingly, we define the **Harmful Rate** as the ratio of harmful errors to total formulas, while the **Mismatch Rate** is calculated as the sum of both invalid and harmful errors divided by the total. The evaluation prompts are illustrated in Figure 17.

---

**Bad Case in LTL Translator**

```
{
  "type": "sudden_repeat",
  "description": "The agent repeatedly moves the same egg (egg 3) to and from the
      fridge without any functional purpose or progress toward heating or storing.
      For example, in successful_traj_2 and traj_4, the agent moves the egg to the
      fridge multiple times after already picking it up, indicating redundant and
      inefficient behavior.",
  "items": ["egg 3", "fridge 1"]
},
Neg LTL Eval Fail: {
  "formula": "G(obs_egg3_state_heated -> G(!act_move_egg3_to_fridge))",
  "is_faithful": false,
  "error_type": "HARMFUL",
  "reason": "The formula incorrectly forbids moving the heated egg to the fridge,
      which is the desired action, thus punishing valid behavior."
}
```

---

*Figure 8.* Bad case found in the evaluation of LTL translator.

We conducted a further analysis of the observed failure cases (Figure 8). We found that the LTL translator struggles to consistently and accurately formalize logical relationships when encountering complex composite logic. (Although we observed instances where the translator autonomously synthesized composite logic in successful cases, this capability proved unstable). Conversely, a naive application of standard formula templates often results in logical errors. These insights provide valuable guidance for future enhancements to the LTL translation module; specifically, we aim to address these complexities through targeted fine-tuning in future work.

## C. Pseudo Code

Algorithm 1 summarizes the full GLARE training procedure. Compared to vanilla GRPO, we highlight the integration of the *Neuro-Symbolic Reward Mechanism*. In particular, the generation of dense rewards is implemented by synthesizing LTL constraints via a one-time holistic group analysis, followed by executing a deterministic automaton for each trace. This design ensures that the computationally expensive reasoning is amortized across the entire group, incurring minimal

marginal cost per step. Furthermore, the automaton-based verification involves only basic state transitions, which are computationally negligible. As such, GLARE preserves the critic-free, low-memory, and stable convergence properties of group-based RL, while introducing mathematically consistent, fine-grained credit assignment that effectively bridges the gap between high-level reasoning and low-level execution.

---

**Algorithm 1** Neuro-Symbolic Reward Shaping via Group-Level Automata (GLARE)

---

 1: **Input:** Dataset $\mathcal{Q}$, Policy $\pi_\theta$, KL coef $\beta$
 2: **for** each iteration **do**
 3:     Sample prompts $q \sim \mathcal{Q}$ and groups $\mathcal{G}_q = \{\tau_1, \ldots, \tau_G\} \sim \pi_\theta(\cdot|q)$
 4:     **for** each group $\mathcal{G}_q$ **do**
 5:         Extract AP sequence for all traces: $\mathbf{L} \leftarrow \phi_{AP}(\mathcal{G}_q)$
 6:         Analyze group and synthesize LTL: $\varphi \leftarrow \pi_{trans}(\text{LLM\_Judge}(\mathcal{G}_q))$
 7:         Compile LTL into Automaton: $\mathcal{A}_\varphi \leftarrow \text{Convert}(\varphi, \mathbf{L})$
 8:         **for** each trajectory $\tau_j \in \mathcal{G}_q$ **do**
 9:             Get AP sequence vector: $\mathbf{l}_j \leftarrow \mathbf{L}[\tau_j]$
10:             Execute Automaton (Vectorized): $\mathbf{r}_{dense,j} \leftarrow \text{Scan}(\mathcal{A}_\varphi, \mathbf{l}_j)$
11:             Total Reward Vector: $\mathbf{R}_j \leftarrow \mathbf{r}_{out,j} + \beta \cdot \mathbf{r}_{dense,j} + \mathbf{r}_{fmt,j}$
12:         **end for**
13:         **for** each step $t = 1 \ldots T_{max}$ **do**
14:             Compute stats: $\mu_t, \sigma_t \leftarrow \text{mean/std}(\{R_{i,t}\}_{i=1}^G)$
15:             **for** each trajectory $\tau_j$ **do**
16:                 $A_{j,t} \leftarrow (R_{j,t} - \mu_t)/(\sigma_t + \epsilon)$
17:             **end for**
18:         **end for**
19:     **end for**
20:     Update $\theta$ via GRPO objective $\mathcal{J}(\theta)$ on batch $\{A_{j,t}\}$
21: **end for**

---

# D. Dynamics of LTL formula

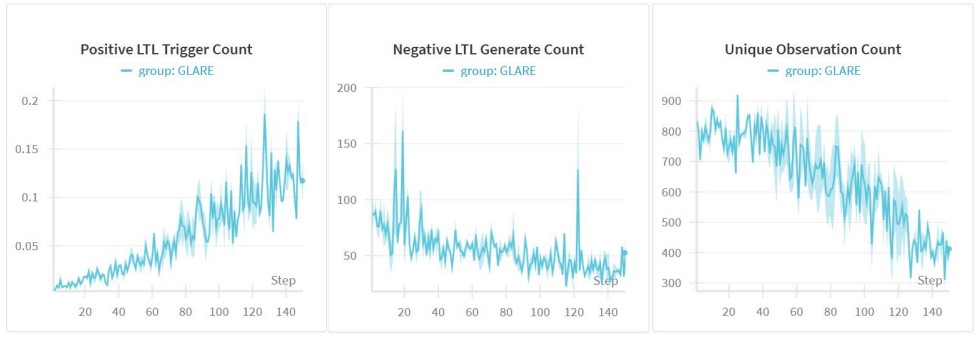

*Figure 9.* Dynamic features of GLARE.

We analyze the learning dynamics induced by GLARE by examining how its reward structure evolves alongside policy improvement and highlight three representative trends (Figure 9) that reveal distinctive properties of our neuro-symbolic reward paradigm. As training progresses, we observe a reduction in the number of unique observations encountered, accompanied by an increase in the activation frequency of milestone events. Beyond binary task success rates, these signals offer richer insights into the agent's intermediate behaviors. More importantly, we find a significant decrease in the number of synthesized negative LTL constraints. In GLARE, negative LTLs are introduced only when novel semantic violations are identified at the group level. As the agent's behavior improves, fewer such violations are detected, and the corresponding penalizing constraints naturally diminish. This adaptive contraction of negative rewards demonstrates that GLARE does not impose arbitrary or persistent penalties, but dynamically adjusts its supervision to reflect the agent's evolving competence.

# E. Effect of Intra-Group Variance on GRPO Advantages

Let $G = \{\tau_1, \ldots, \tau_k\}$ be a group of trajectories sampled from policy $\pi_\theta$, with latent ground-truth rewards $r^*(\tau_i)$.

GRPO computes advantages as:

$$A_i = \frac{r(\tau_i) - \bar{r}}{\sigma_{adv} + \delta}, \quad \bar{r} = \frac{1}{k} \sum_{j=1}^{k} r(\tau_j), \tag{8}$$

where $\sigma_{adv} + \delta > 0$. Thus, the relative ordering of advantages depends only on the centered reward term $r(\tau_i) - \bar{r}$.

**Lemma E.1** (Translation Invariance)**.** *Let the reward proxy satisfy:*

$$r(\tau_i) = r^*(\tau_i) + b + \eta_i, \tag{9}$$

*where $b$ is a constant systematic bias shared across the group and $\eta_i$ is an arbitrary error term. Then the group-centered rewards satisfy:*

$$r(\tau_i) - \bar{r} = (r^*(\tau_i) - \bar{r}^*) + (\eta_i - \bar{\eta}), \tag{10}$$

*and the constant bias $b$ has no effect on the ordering of $\{A_i\}$.*

*Proof.* Substituting the reward proxy definition into the centering equation:

$$\begin{aligned} r(\tau_i) - \bar{r} &= (r^*(\tau_i) + b + \eta_i) - \frac{1}{k} \sum_{j=1}^{k} (r^*(\tau_j) + b + \eta_j) \\ &= (r^*(\tau_i) + b + \eta_i) - (\bar{r}^* + b + \bar{\eta}) \\ &= (r^*(\tau_i) - \bar{r}^*) + (\eta_i - \bar{\eta}). \end{aligned}$$

The constant term $b$ cancels out explicitly. Thus, if $\eta_i = 0$ for all $i$, the estimated advantage order is identical to the ground-truth advantage order. $\square$

**Lemma E.2** (Sensitivity to Intra-Group Variance)**.** *Assume there exist two trajectories $\tau_i, \tau_j \in G$ such that $r^*(\tau_i) > r^*(\tau_j)$. If the error terms $\eta_i$ and $\eta_j$ are not almost surely equal, then the probability of a **Rank Reversal** (i.e., $A_i < A_j$) is strictly positive.*

*Proof.* A rank reversal occurs whenever the estimated advantage ordering contradicts the ground truth:

$$A_i < A_j \iff r(\tau_i) - \bar{r} < r(\tau_j) - \bar{r} \iff r(\tau_i) < r(\tau_j). \tag{11}$$

Substituting the proxy definition:

$$\begin{aligned} r^*(\tau_i) + b + \eta_i &< r^*(\tau_j) + b + \eta_j \\ r^*(\tau_i) - r^*(\tau_j) &< \eta_j - \eta_i. \end{aligned}$$

Let $\Delta^* = r^*(\tau_i) - r^*(\tau_j) > 0$. The condition for rank reversal becomes: $(\eta_j - \eta_i) > \Delta^*$.

For any stochastic error distribution with non-zero intra-group variance (e.g., $\eta \sim \mathcal{N}(0, \sigma^2)$), the probability $P(\eta_j - \eta_i > \Delta^*)$ is strictly positive. $\square$

**Corollary E.3** (Consistency Regime)**.** *If $\eta_i = \eta_j$ almost surely for all $(i, j)$ in the group (i.e., zero intra-group variance), then GRPO preserves the ground-truth ranking exactly. Otherwise, ranking errors occur with positive probability.*

**Remark.** Lemma 1 implies that **systematic, group-wise bias** ($b$) is benign under GRPO due to the translation invariance of the advantage function. In contrast, Lemma 2 shows that **intra-group variance** ($\eta$) directly destabilizes the advantage ordering, independent of the bias magnitude. Therefore, reward proxies that induce low or correlated intra-group error (such as GLARE's deterministic automata) yield theoretically more stable optimization signals than unbiased but stochastic evaluators.

# F. Prompts

In this section, we present the comprehensive set of prompts utilized throughout the GLARE pipeline, accompanied by a brief functional overview of each component: **Triplet Extractor (Figure 10):** This prompt guides the extraction of subject-predicate-object triplets from natural language observations in a fixed format. These triplets serve as the foundation for generating open-semantic Atomic Propositions (APs). **Trajectory Analyst (Figure 11 and Figure 12):** This module performs a dual analysis of the trajectory group. It is responsible for synthesizing potential milestones with their temporal dependencies from successful traces, while simultaneously detecting and classifying bad behaviors from failed traces. For clarity, we display the Chain-of-Thought (CoT) reasoning process in Figure 12 and the final structured classification in Figure 11. **LTL Translators (Figure 13 and Figure 14):** These prompts facilitate the translation of the analyzed milestones, classified bad behaviors, and extracted APs into executable LTL formulas. We provide separate prompts for Positive and Negative LTL generation, both of which adhere to template-based construction. **Evaluator (Figure 15,Figure 16 and Figure 17):** These prompts are used to quantitatively evaluate the sub-modules introduced by GLARE.

---

**Prompt Template for Triplets ( used for APs ) Extractor**

You are a precise Semantic State Extractor. Your task is to analyze the Current Observation (and Goal) to extract structured state triplets representing the world state.
Output Format (JSON):

```
{"thought": "Reasoning about visible objects, attributes, and goal relevance.",
    "triplets": [
        ["Subject", "Category", "Value"],
        ...]}
```

[DEFINITIONS]
- Subject: The Core Entity ID (e.g., "apple 2", "drawer 1", "self"). NO adjectives.
- Category: The attribute type (MUST be one of: "loc", "state", "goal_match","seen").
- Value: The state content, it is single word in most of time(e.g., "true", "apple 1", "closed", "cool").
[STRICT RULES - READ CAREFULLY]
1. **Subject Normalization**: Use underscores for IDs. Subject must be the NOUN only.
- BAD: ["unsliced tomato 1", "loc", "table 2"] - GOOD: ["tomato 1", "loc", "table 2"], ["tomato 1", "state", "unsliced"]
2. **Inventory Rule (Item-Centric)**: If agent holds/pick/get X, the location of X is 'self'.
- BAD: ["self", "holding", "apple 1"] - GOOD: ["apple 1", "loc", "self"]
3. **Self Location**: 'self' location must be a ROOM or REGION.
- BAD: ["self", "loc", "holding apple 2"] - GOOD: ["self", "loc", "kitchen 2"]
4. **seen Attribute**: Indicate if an item is visible.
GOOD: ["apple 1", "seen", "true"]
5. **No Hallucination**: Only extract explicitly visible facts. Do NOT extract the agent's action (that is handled externally).
[EXAMPLES]
Input: "You are in the middle of a room. You see a closed drawer 1 and an unsliced tomato 1 on the table 2."
Output:

```
{"thought": "You are in a room, the 'middle of' is needless. You pick up cup 2.
    Drawer 1 is visible and closed. Tomato 1 is visible, unsliced, and on the table
    2.",
    "triplets": [
        ["self", "loc", "room"],
        ["cup 2", "loc", "self"],
        ["drawer 1", "state", "closed"],
        ["tomato 1", "state", "unsliced"],
        ["tomato 1", "loc", "table 2"]
    ]}
```

*Figure 10.* The prompt template of triplets extractor.

**Prompt Template for Group Trajectory Analyst (Critic)**

You are an agent trajectory analysis expert.

Your task is to analyze multiple interaction trajectories corresponding to the same task, identify the agent's Inefficient Behavior Patterns and milestones,

## RULES

1) Ensure that all identified issues can be expressed using temporal logic.

2) Clearly specify which concrete actions and which related objects/items are involved in each identified issue. Provide clear and comprehensive information for downstream processes.

3) When a single issue involves multiple similar logical structures or similar items, they may be merged in description. However, you must strictly comply with Rule 2: all relevant details must remain complete, and no concrete information should be lost due to over-merging.

4) Milestone Extraction (Must strictly follow):

**State-driven verification**:

A Milestone must correspond to an objectively occurring physical state change, such as an Inventory Change or Object State Change.

**Strict "completed" criterion**:

If there are successful trajectories: directly extract the key nodes along the successful path. If all trajectories are failures: only extract steps that are both necessary for the task goal and have actually been fully executed in the trajectory. No inference allowed: events that did not actually occur must never be labeled as Milestones.

**Empty-set principle**:

If the trajectory contains only invalid information (e.g., navigation, idle looping), directly return an empty list []. Do not force milestones, and do not fabricate data just to "complete the task".

**Minimal necessary event set principle**:

Even successful trajectories contain redundant information. You must strictly filter out only the true Milestones, removing irrelevant details. Milestones should form a minimal necessary event set, not a long or verbose event sequence.

5) Bad Behavior Patterns:

When identifying Bad Behavior Patterns, do not restrict the agent's exploration. We only restrict behaviors that are inefficient, ineffective, or unsafe.

Do not describe problems in terms of "what the agent did not do". Instead, specify "under what conditions the agent did what action, leading to a problem."

For example: Bad: "The agent failed to obtain item A." Good: "The agent needed item A, saw item A, but did not pick it up."

You must classify each Bad Behavior into one and only one of the following common categories. Note: A single category may contain multiple different Bad Behaviors and avoid duplication or omission.

**global_repeat:** Repeatedly performing meaningless actions. For example, revisiting a location or inspecting an object that cannot possibly change, after it has already been checked. Repeating an action sequence (A → B → A → B) is also typically meaningless, such as repeatedly browsing static areas (A → B → C → C → A → B → C).

**sudden_repeat:** Immediately repeating the same action consecutively without sufficient justification. For example, performing action A and then immediately performing action A again is generally abnormal.

**lack_premise:** Executing an action without satisfying its prerequisite conditions. For example, action B only makes sense when item A is available, but the agent performs B prematurely. NOTICE: Exploratory behaviors do not have prerequisites and are not subject to this rule—be careful to distinguish them.

**hazard:** Irrational or potentially dangerous behaviors.

**rollback:** Regressing task progress. For example, after acquiring a task-required item, discarding it before using it correctly—this is generally abnormal.

**other:** Special issues that are difficult to classify into the above categories.

*Figure 11.* The prompt template of Trajectory Analyst (Critic).

---

**Prompt Template for CoT Guide and Output Example**

##FORMATS
Please strictly follow the format below when outputting your analysis results:
```
<think>
```
# step 1: Analyze the task
# step 2: From all trajectories, based on Rule 4, identify content in the current trajectory that clearly qualifies as milestones, and specify all concrete items involved after filtering
# step 3: Identify the temporal relationships among the selected milestones and split them accordingly
# step 4: Check: "If I completely remove this sub-milestone, would the task not only fail but become physically impossible to complete?"
# step 5: Identify all Bad Behavior Patterns in the trajectory, based on Rule 5
# step 6: Classify the bad behaviors and specify the concrete items involved; ensure all relevant items are included, not just partial examples
# step 7: Check whether the detailed descriptions of bad behaviors (not limited to type labels) are duplicated or highly similar; if so, merge them. Check whether the items listed actually exist in the trajectory; if not, replace or remove them. Check whether all relevant items are correctly referenced; if not, add them immediately.
```
</think>
```

```json
\{
  "milestones": [
    \{
      "tag": 1,
      "description": "A concrete and explicit description of milestone 1",
      "items": ["item1", "item2"],
      "dependency": []
      // If this milestone depends on previous milestones, list their tags here, e.g
          . [1, 2].
      // If there is no dependency, leave it as an empty list [].
    \},
    \{
      "tag": 2,
      "description": "A concrete and explicit description of milestone 2",
      "items": ["item3"],
      "dependency": [1]
    \}
    ...
  ],
  "bad\_behaviors": [
    \{
      "type": "Bad behavior category A",
      "description": "A detailed explanation of failure/inefficiency issue 1",
      "items": "All objects, items, or entities involved in this issue"
    \},
    \{
      "type": "Bad behavior category A",
      // Same category as above, but involving a different target or context
      "description": "A detailed explanation of failure/inefficiency issue 2",
      "items": "All objects, items, or entities involved in this issue"
    \},
    \{
      "type": "Bad behavior category B",
      // A different type of issue
      "description": "A detailed explanation of failure/inefficiency issue 3",
      "items": "All objects, items, or entities involved in this issue"
    \}...]\}
```

*Figure 12.* The chain of thought guideline and few-shot for critic.

**Prompt Template for Positive LTL formula translator**

```
You are a professional NL2LTL translator, you need to translate the success analysis
    of agent trajectories into standard LTL statements based on the given AP list.
AP list is the APs that appear in the trajectory, not a complete world model

##GUIDELINES
**Translate sub-milestones**
Select the APs corresponding to each sub-goal and connect them using basic logical
    operators & and | to construct the LTL expression for each sub-goal.
If any AP in need is missing from the AP list, treat that AP as 'False'. 
 E.g.,
    You need 'G(obs_A -> (obs_B & obs_C))', but can not find obs_B, it becomes 'G(
    obs_A -> (False & obs_C))' which implies '!obs_A'.

**Temporal Composition**
Logic: According to the temporal relationships between sub-milestones, construct a
    Strict Sequence using Nested 'F'.
1.For milestones with a clear sequential order, use nested F operators to express a
    strict temporal sequence, for example: 'F(Step1 & F(Step2 & F(Step3...)))'.
2.For parallel cases without an obvious temporal order, use
F((A & F(B)) | (B & F(A)))
to provide an accurate logical representation.

##NOTE
According to the required temporal relationships, check whether parentheses are
    correctly used in each layer of nesting within the current LTL formula.
Ensure that all APs used appear in the AP list; if not, replace them with other APs
    or with False.

##OUTPUT FORMAT
<think>
1. Identify the sub-milestones and check their logic relation (& or |). Then
    translate them into LTL formula following the ##GUIDELINES. Retaining only the
    most critical information to avoid over-description.
2. Construct all the sub-pos_ltl formula into SINGLE formula following the ##
    GUIDELINES.
3. Ask yourself: "Does all logical relations and temporal relations be correctly
    expressed in the formula with right symbols?"
4. Recheck the candidate rules following the ##NOTE step by step.
</think>

<pos_ltl>
...
</pos_ltl>
```

*Figure 13.* The prompt template of positive ltl formula translator.

**Prompt Template for Negative LTL formula translator**

```
You are a professional NL2LTL translator, you need to translate the failure analysis
    of agent trajectories into standard LTL statements based on the given AP list.
AP list is the APs that appear in the trajectory, not a complete world model

##GUIDELINES
**Critical:** If any AP in need is missing from the AP list, treat that AP as 'False
    '. 
 E.g., You need 'G(act_A -> (obs_B & obs_C))', but can not find obs_B, it
    becomes 'G(act_A -> (False & obs_C))' which implies '!act_A'.

According to inputs' 'type' select LTL template. Based on the above basic syntax
    reference, you can also use '&' and '|' to flexibly combine LTL formulas.
- global_repeat: Never do it again. 'G(act_A -> X(G(!act_A)))'
- sudden_repeat: Not repeat it in sudden. 'G(act_A -> X(!act_A))'
- lack_premise: Missing premise for action. G(!obs_A -> !act_B)
- hazard: Something seems dangerous. 'G(!act_Subject)'
- rollback: Progress rollback. 'G(obs_Trigger -> G(!act_Forbidden))'
- other: Other special cases, use your judgment to construct the LTL formula.

##NOTE
- Not to write multiple actions happening simultaneously, such as (act_A & act_B),
    since actions are instantaneous and agents can only perform one action at a time,
     you need to deal with it one by one.
- When you need to output many LTL constraints that are logically similar but
    involve different subjects, do not merge them into a single complex formula.
    Instead, split them into multiple independent formulas, with each formula
    focusing on one specific constraint.
- Split indenpendent LTL formula with \n. Do not make single heavy formula.
- Only output clean LTL formula without extra explanation in <neg_ltl>.
##OUTPUT FORMAT
<think>
1.Please identify ALL the bad behaviors and their templates (restate the template),
    then construct the <neg_ltl>formula following the ##GUIDELINES 2.
2.Ensure do not miss bad behaviors or items, but ensure never use unexistent APs.
2.Recheck the candidate rules following the ##NOTE step by step.
</think>

<neg_ltl>
...
</neg_ltl>
```

*Figure 14.* The prompt template of negative ltl formula translator.

**Prompt Template for Atomic Proposition (AP) Extractor Evaluation (Redundancy Eval)**

```
You are an expert Logician and Semantics Analyst.
Your task is to check a list of "Atomic Propositions" (APs) for **Redundancy**.
We want to ensure that the AP list for a task group is concise and distinct.

[DEFINITION OF REDUNDANCY]
Two APs are redundant if they assert the same atomic fact about the world, even if
    the wording or syntax differs slightly.
Since APs encompass system states at different moments, it is normal for an object
    to have multiple attributes, such as being in different locations. However, if
    two APs express the same attribute or state, they are considered redundant.

- Redundant: "mug state clean" vs "mug state cleaned"
- Redundant: "egg state heat" vs "apple state hot"
- NOT Redundant: "door state closed" vs "door state open" (Different attributes)
- NOT Redundant: "apple 1 loc table 1" vs "apple 2 loc table 1" (Different subjects)

[INPUT GROUPS]
{batch_input}

[OUTPUT INSTRUCTION]
Return a JSON object with a "results" list.
Format for each result:
{{
    "reason": "Brief explanation of the overlap.",
    "has_redundancy": true/false,
    "redundant_sets": [
        ["ap1", "ap2"] // Lists of strings that are assert same world fact.
    ],"
}}
Output JSON only.
```

*Figure 15.* The prompt template of Atomic Proposition (AP) Extractor Redundancy Evaluation.

**Prompt Template for Atomic Proposition (AP) Extractor Evaluation (Accuracy&CompletenessEval)**

```
You are an expert Evaluator for a Neuro-Symbolic State Extractor.
Your task is to evaluate the **Accuracy** and **Completeness** of extracted "Visual
    Triplets" for a given Observation and Goal.

[EXTRACTION RULES - THE STANDARD OF TRUTH]
The system SHOULD have followed these rules:
1. **Format**: `["Subject", "Category", "Value"]`
2. **Inventory**: If agent holds X, output `["X", "loc", "self"]`. (NOT `["self", "
    holding", "X"]`).
3. **Self Location**: `["self", "loc", "room_name"]`.
4. **No Hallucination**: Only extract explicitly visible facts. But only ['xxx', '
    seen', 'false'] is allowed.

[EVALUATION CRITERIA]
1. **Accuracy**: Are the extracted triplets correct according to the Observation and
     Rules?
   - Check for hallucinated objects.
   - Check for wrong attributes (e.g., saying "open" when it says "closed").
   - Check for rule violations (e.g., "holding" relation instead of "loc" "self").
2. **Completeness**: Are there missing important facts?
   - Any visible object in the observation NOT mentioned in triplets?
   - Any important state (open/closed, sliced/cooked) missing?

[INPUT CASES]
{batch_input}

[OUTPUT INSTRUCTION]
Return a JSON object with a "results" list.
Format for each result:
{{
    "case_id": <int>,
    "bad_triplets": [0, 2], // Indices (0-based) of triplets that are incorrect,
        invalid format, or hallucinations. Empty list if all perfect.
    "has_omission": true/false, // True if important facts are missing.
    "missed_facts": [["Subject1", "Category1", "Value1"],["Subject2", "Category2", "
        Value2"]], //Brief list of missing items/states.
    "reason": "Explanation..."
}}
Output JSON only.
```

*Figure 16.* The prompt template of Atomic Proposition (AP) Extractor Accuracy&Completeness Evaluation.

---

**Prompt Template for LTL Translator Evaluation (Eval)**

You are a Formal Logic Expert. Your task is to verify the **Translation Quality** of LTL Formulas based on Trace Analysis.

**[LTL GRAMMAR REFERENCE]**

```
- sudden_repeat: 'G(act_A -> X(!act_A))'
- lack_premise: 'G(!obs_A -> !act_B)'
- hazard: 'G(!act_Subject)'
- rollback: 'G(obs_Trigger -> G(!act_Forbidden))'
- other: Flexible combination using '&', '|', '!', 'X', 'F', 'G', 'U'.
```

**[CONTEXT]**
**Positive LTL** corresponds to successful milestones.
**Negative LTLs** correspond to different failure modes (bad behaviors). These formulas represent \*\*independent\*\* tracks of behavior. A formula is **Faithful** if it correctly translates the text in the Analysis JSON into LTL.
**[EVALUATION CRITERIA]**
1. **Positive LTL**: If faithful, set 'is_faithful' to true. If NOT faithful, classify the error type: **LOGIC_ERROR**: The formula enforces a WRONG sequence or logic that contradicts the analysis (e.g., wrong order, wrong action). This is CRITICAL. **MISSING_INFO**: The formula is logically correct but skips some steps mentioned in the analysis. This is MINOR.
2. **Negative LTLs**: Evaluate EACH formula. First, identify which bad behavior this formula corresponds to. If faithful, set 'is_faithful' to true. If NOT faithful, classify the error type: **HARMFUL**: The formula is wrong in a way that would punish VALID behavior (e.g., forbidding a necessary action). This is CRITICAL. **INEFFECTIVE**: The formula fails to capture the bad behavior (e.g., syntax error, irrelevant constraints) but likely won't punish good behavior actively. It's a "miss".
**[INPUT CASES]**
{ batch_input }
**[OUTPUT INSTRUCTION]**
Return a thinking process within `<think>` tag and a JSON object with a "results" list.
For THINKING:
`<think>`
1. Examine the Positive LTL: Does it capture the sequence of milestones? If there's an error, is it a complete logic failure (LOGIC_ERROR) or just a skip of some steps (MISSING_INFO)?
2. Examine the Negative LTL formulas one by one: - Attempt to match each formula with the existing bad behaviors from the analysis, selecting the one it most likely belongs to. - Based on this alignment, judge whether the formula faithfully reflects the content and type of that bad behavior. - If it is not faithful, decide the impact: Would it punish valid actions (HARMFUL) or is it just a brokenuseless formula (INEFFECTIVE)?
`</think>`
For the JSON:

```
Each entry in "results" must strictly follow this structure:
{{
    "case_id": <int>,
    "pos_ltl_eval": {{ "is_faithful": true/false, "error_type": "NONE" | "
        LOGIC_ERROR" | "MISSING_INFO", "reason": "..." }},
    "neg_ltl_evals": [
        {{ "formula": "...", "is_faithful": true/false, "error_type": "NONE" | "
            HARMFUL" | "INEFFECTIVE", "reason": "..." }},
        ...
    ]
}}
```

Output JSON only.

*Figure 17.* The prompt template of LTL Translator Evaluation.

