# OpenReview forum: "GLARE: Scalable Neuro-Symbolic Reward Shaping for LLM Agents via Group-Level Automata"
_ICML.cc/2026/Conference — ICML 2026 regular_

### Official Review · Reviewer_VpUZ · 2026-03-09

**Soundness:** 2
**Presentation:** 3
**Significance:** 4
**Originality:** 4
**Overall Recommendation:** 4
**Confidence:** 4

**Summary:**

The paper introduces GLARE, a neuro-symbolic reward shaping framework designed to address the sparse reward problem in GRPO for LLM agents engaged in long-horizon tasks. Existing LLM-as-a-judge approaches suffer from high computational costs, reasoning-answer misalignment, and intra-group inconsistencies that degrade relative advantage estimation. GLARE tackles this by decoupling semantic understanding from credit assignment. It first uses a lightweight LLM to extract a symbolized state track (APs) from agent trajectories. A critic LLM then analyzes the group of trajectories to identify milestones and bad behaviors, translating them into LTL formulas. Finally, these formulas are compiled into a deterministic FSA that assigns consistent, dense process rewards to the group. Evaluated on ALFWorld using Qwen2.5 models, GLARE outperforms both standard GRPO and conventional trace-level LLM judges, achieving higher success rates while using only 15% of the completion tokens required by trace-level evaluators.

**Compliance With Llm Reviewing Policy:**

Affirmed.

**Key Questions For Authors:**

1. How adaptable is the Triplet Extraction schema to environments that are not object-centric or spatial (e.g., mathematics or code debugging)? Would the categories (loc, state, seen) need to be completely rewritten, and if so, how much manual engineering is required per new environment?
2. In Section 6.3, you mention a 12% Mismatch Rate for LTL translation. When a formula fails to compile or is deemed invalid, what is the exact fallback mechanism during that specific GRPO update step?
3. Are there scenarios where the global group analysis hallucinates a "milestone" that is structurally invalid but syntactically correct, thereby misguiding the entire group? If so, how often did this occur?

**Limitations:**

Yes

**Strengths And Weaknesses:**

# Strengths

* **Originality:** The neuro-symbolic approach of using LLMs to synthesize temporal logic rules, which are then evaluated deterministically via automata for RL credit assignment, is highly innovative. Decoupling the semantic parsing from the scoring mechanism is an elegant solution to the reasoning-answer drift common in LLM judges.
* **Significance:** The high computational cost and variance of LLM-based reward models are major bottlenecks in agentic RL. Reducing the token footprint to 15% of the trace-level judge baseline while simultaneously improving performance represents a highly practical and significant contribution to the field.
* **Soundness (Theoretical):** The theoretical justification provided in Appendix E (Lemma E.2 and Corollary E.3) is exceptionally strong. Proving that intra-group variance directly destabilizes the advantage ordering in GRPO provides a rigorous mathematical foundation for why deterministic automaton-based evaluation is superior to stochastic LLM scoring.
* **Presentation & Transparency:** The paper is well-structured and clearly written. The authors are commendably transparent about the failure modes of their sub-modules (e.g., AP Extractor omissions and LTL Translator mismatch rates in Section 6.3 and Appendix B).

# Weaknesses

* **Soundness (Empirical Breadth):** The most significant weakness is the reliance on a single benchmark (ALFWorld). For an ICML paper proposing a "scalable" and general RL framework for LLM agents, evaluating on only one text-based environment is insufficient. ALFWorld's highly structured textual observations artificially favor the Triplet Extractor's schema (Subject, Category, Value). It is unclear how robustly this framework translates to domains with vastly different observation spaces (e.g., web navigation like WebArena/WebShop, or coding environments like SWE-bench).
* **Methodology:** The AP extraction relies heavily on few-shot prompting and predefined categories (loc, state, goal\_match, seen). This rigid schema might bottleneck the agent in more open-ended environments where state changes cannot be easily boxed into these specific attributes.
* **Presentation:** While generally strong, the main text leaves the theoretical motivation (Appendix E) entirely to the appendix. Briefly bringing the core intuition of Lemma E.2 into Section 6.1 would greatly strengthen the paper's narrative.

# Suggested improvements
* **Additional Environment:** Evaluate GLARE on at least one additional benchmark outside of the embodied text domain (e.g., WebShop, WebArena, or an interactive coding task) to demonstrate the generalizability of the Triplet Extractor and LTL Translator.
* **Ablation on Group Size:** GRPO's performance is sensitive to the group size. It would be valuable to see an ablation showing how GLARE scales with different sizes compared to baseline LLM judges.
* **Analysis of LLM capabilities:** The framework utilizes Qwen3-4B-Instruct for extraction and criticism. How sensitive is the final policy performance to the capability of this specific reward-generating LLM? A brief comparison using a weaker or stronger model would add depth.

---

> ### Author Rebuttal · Authors · 2026-03-31
>
> We are grateful for the constructive review. Below, we provide detailed clarifications to the highlighted issues.
>
> **Q1&W1&W2**.
> The Triplet extraction schema is completely oriented towards object-centric domains, as it can effectively perform state tracking for various objects in the environment. We also conducted experiments on WebShop and experimental results is above.
>
> |Method| Score | Succ.|
> |-|-|-|
> |GRPO|75.8|56.8|
> |LLM Judge|78.8|61.2|
> |GLARE|80.2|63.4|
>
> Although the structure of the observation space in WebShop is different, the AP Extractor remains applicable to WebShop—which is also object-centric—after updating few-shot examples and relaxing category generation. Predefining 'categories' in ALFWorld was intended to ensure consistency during AP extraction; however, when facing new environments, the schema can be adapted by expanding categories as needed. It is not incapable of handling "environments where state changes cannot be easily boxed into these specific attributes." In WebShop, because the required categories are already embedded in the observations (such as product 'size', 'color', etc.), consistency can be maintained even when the generation is fully open. The prompts for the LTL Translator and Critic do not even rely on domain-specific few-shot examples and are similarly transferable. However, it is difficult to abstract "entities" and "events" in math and code domains, and there are rarely temporal relationships between events; therefore, we acknowledge they are not suitable for our current GLARE framework. The term "scalable and general RL framework" indeed has its limitations, and we will explicitly discuss this point in the revised version.
>
> **Q2**. The specific fallback mechanism is as follows: when a formula is invalid, the relative advantage estimation for the trajectories in this specific group will revert to the standard outcome-reward-only GRPO. This will not interfere with the utilization of process rewards for data in other groups; it merely wastes a small amount of the judge's tokens, whereas successful formula generations consistently yield significant performance improvements.
>
> **Q3**.
> The phenomenon of hallucinating a 'milestone' does exist. In our prompts, we specifically emphasize that the critic must base its extraction on existing trajectories and avoid over-imagining to prevent potentially erroneous guidance. By doing so, we have not observed the generation of completely non-existent milestones. However, during a human evaluation of 50 sampled formulas, we found that meaningless sub-milestones still exist (in 6 samples). For example, in a cool lettuce task, a meaningless state (obs_lettuce1_loc_countertop1) in the successful trajectories was mistakenly extracted as a milestone. Upon further observation of the group trajectories, this occurred because there were few successful samples (1 or 2) and all of them happened to exhibit this action. The agent's behavior within this specific group misled the critic, which indeed causes slight misguidance. But this is merely an occasional phenomenon and typically only appears at a specific step. Once more successful trajectories emerge later (especially those that bypass the meaningless 'milestone' but still succeed), due to the dynamic updating of LTL formulas, the critic will no longer consider this invalid sub-milestone.
>
>
> **Presentation**:
> We greatly appreciate the reviewer's suggestion to bring the core intuition of Lemma E.2 forward, and we will adopt this suggestion in the final version.
>
>
> **Ablation**:
> We have supplemented ablation experiment results for $n=4$ and using another reward model. As can be seen, although the overall performance under the $n=4$ setting is significantly weaker than $n=8$ given the same amount of training, GLARE's advantage over the LLM-as-a-Judge baseline remains obvious. Under the condition where the judge model is replaced with Qwen3-30B-A3B-Instruct, our conclusions still hold.
>
> |method | LLM judge | GLARE|
> |-|-|-|
> |baseline|76.4|84.5|
> |rollout_n=4|62.2 | 71.4 |
> |Qwen3-30B-A3B-Instruct|78.2|85.7|

---

### Official Review · Reviewer_z6zx · 2026-03-11

**Soundness:** 3
**Presentation:** 2
**Significance:** 3
**Originality:** 3
**Overall Recommendation:** 4
**Confidence:** 4

**Summary:**

This paper proposes a framework that uses Linear Temporal Logic (LTL) and finite state automata to provide reward signals for training LLM agents combined with Group Relative Policy Optimization (GRPO). The method first extracts atomic propositions (APs) from trajectories using LLMs, translates them into LTL formulas, and compiles these into deterministic automata that track agent progress. The approach achieves 12.1% improvement over GRPO and 8.1% over LLM-as-a-Judge baselines on ALFWorld while using only 15% of the computational cost.

**Compliance With Llm Reviewing Policy:**

Affirmed.

**Final Justification:**

The core contribution, integrating symbolic LTL logics with RL and LLM agents for dynamic reward specification, is valuable, even with missing citation and clarification between existing works. Also, this is well-supported by strong empirical results: 12.1% improvement over GRPO and 8.1% over LLM-as-a-Judge on ALFWorld.

My main concerns was full automation and novelty over claim on reward shaping. Even though the reward shaping is not fundamentally richer than prior automaton-based progress-plus-penalty methods, the main novelty lies in LLM-driven dynamic generation of diverse logical events, not the reward form itself, as authors acknolwedged. During revision, this should become clear comparing the proposed one with the existing method. The authors have adequately addressed my concerns. With proper revisions to the reward claims, I now support acceptance.

**Key Questions For Authors:**

1. How many automaton states exist for each task, and how dense it is compared to acceptance state only LTL reward.
2. The reward coefficients (pos=2.0, neg=-0.5, trend=0.5) seems to be manually tuned. How sensitive is performance to these values?
3. Why do authors use triplets (subject, category, value)  for atomic proposition extraction? It does not have any justification for this design choice.
4. following 3, would binary relations (subject, predicate, object) be more expressive? Would hypergraphs capture more state information?
5. Is there any study on top of this automating everything like [1]?
6. Regarding reward shaping, it does look like the paper [2] has more various reward design choice including hybrid reward, have you tried the technique?
7. The negative-only ablation performs worse than baseline. This suggests that purely punitive signals are harmful. How should practitioners balance positive and negative LTL, and does this balance vary by task type?

I am currently leaning weak reject, but I think the core idea has merit. In the author response, additional justification of the triplet selection, improved coverage of related work (automated procedures [1] or other reward design [2]), and responses to questionswould help me re-evaluate this paper.

[1] ARM-FM: Automated Reward Machines via Foundation Models for Compositional Reinforcement Learning, Roger Creus Castanyer, Faisal Mohamed, Pablo Samuel Castro, Cyrus Neary, Glen Berseth

[2] Adaptive Reward Design for Reinforcement Learning, Minjae Kwon, Ingy ElSayed-Aly, Lu Feng

**Limitations:**

The paper could be strengthened by including a discussion of its limitations, such as the reliance on manually designed prompts/categories and the lack of full automation in the approach like existing work. Additionally, the reward functions appear quite similar to those in some prior works (which could be cited for better context), and while they provide a denser signal than typical LTL-based shaping, they are not fully dense because LTL states often very small. I think this aspect would benefit from explicit acknowledgment. Finally, it would be helpful to see (or at least discuss) sensitivity analysis on key hyperparameters, such as the reward coefficients.

**Strengths And Weaknesses:**

**Strengths**
1. The paper presents a hybrid approach combining the semantic understanding of LLMs with the determinism of formal logic, especially with LTL. These approach similarly has been studied [1] (this work essentially add rigorous temporal information during training through LTL extraction, so fundamentally, sort of reward machine works), but this paper rigorously engineer this idea to combine with GRPO.

2. The method demonstrates consistent improvements across multiple task types in ALFWorld, with particularly notable gains on long-horizon tasks, which are expected from existing LTL related results showing improvements on temporal dependency tasks (Pick: 94.7%, Clean: 87.9%, Heat: 88.2%). The computational efficiency (15% of LLM judge cost) is a practical advantage.

**Weaknesses**
1. Some of missing citations, especially assigning reward signals based on linear temporal logic. They (Equation 6, 7) are essentially providing very much similar reward [2] , without citations. This proves lack of researching prior works.
For example, (i) Milestone rewards (presented in paper) are given when automaton state transitions occur (Eq. 6: $I(q_{t+1} ∈ P(q_t))$) (ii) Trend rewards provide shaping within intervals between milestones. These are the same as the paper [2]'s naive reward.

2. Over-claim: The paper claims to provide "dense" reward signals, but this characterization is misleading. While the rewards are denser than standard LTL-based approaches (which typically only give rewards at acceptance states), GLARE's rewards are still fundamentally sparse.
Because as noted in Weakness 1, it only happens mainly when state transitions happens. But automaton's transitions usually very small.

3. LTL Generation is Not Fully Automated unlike existing work [1]. The LTL generation process has several manual components that are not clearly acknowledged. For example, the paper relies on manually defined categories (sudden_repeat, global_repeat, lack_premise, hazard, rollback, other) with fixed LTL templates. This limits generalizability to novel task domains where failure modes are not known a priori. Also, the triplet extraction and LTL translation heavily depend on carefully designed prompts (shown in Appendix F). The paper does not discuss the effort required to adapt these prompts to new environments.

[1] ARM-FM: Automated Reward Machines via Foundation Models for Compositional Reinforcement Learning, Roger Creus Castanyer, Faisal Mohamed, Pablo Samuel Castro, Cyrus Neary, Glen Berseth

[2] Adaptive Reward Design for Reinforcement Learning, Minjae Kwon, Ingy ElSayed-Aly, Lu Feng

---

> ### Author Rebuttal · Authors · 2026-03-31
>
> We thank the reviewer and clarify the raised concerns.
>
> **Q1&W2**. We generate multiple LTL formulas(1 pos + 0-10 neg per task) to construct the reward. Single negative LTL typically involve 2-5 APs, yielding 2-3 automaton states (computed via ltl2ba). For positive LTL, their automaton state counts depend on sub-milestone temporal phases (approx 2-6).
>
> Regarding the definition of "dense," the focus should be on the proportion of steps within the agent's trajectory that receive additional rewards. This ratio is exceptionally high: even factoring in LTL compilation failures, approx 20% of the steps are assigned extra rewards. This is because a single LTL can trigger multiple times per trajectory, and each trajectory is governed by multiple LTL formulas (one positive LTL and several negative LTLs).
>
> **Q2**. We set reward coefficients intuitively without exhaustive hyperparameter tuning (expensive for LLM training), and our preliminary experiments with penalties (-0.5, -1.0) yielded similar performance, suggesting that GLARE is insensitive to reward coefficients. Our reward design requires only that success > pos > -neg ≥ trend. This ensures outcome reward dominates, milestone rewards provide guidance, and negative penalties remain relatively small (yet not offset by trend reward).
>
> **Q3,4**. Binary relations`(subject, predicate, object)`, describe the **Action Tracking**, which we replace with the raw environmental actions. What we fundamentally require is **State Tracking**. Crucial environmental information manifests as persistent object states, while actions are too transient to anchor temporal relationships. For instance, to enforce "getting coffee requires picking up a cup," an action-based constraint like `G(!self_get_coffee -> self_pick_cup)` fails if the agent drops the cup midway. Conversely, the `(Subject, Category, Value)` schema uses triplets to represent diverse persistent object states (e.g., cup_loc_self), enabling the constraint `G(!cup_loc_self -> self_get_coffee)`.
>
> Triplets also perfectly aligns with the overwrite-and-update logic of a nested dictionary (like Dict[subject[category]]), ensuring the uniqueness and Markovian property of the state (<'door','state','open'> will be merged by <'door','state','close'> in future). Furthermore, the cost and capability requirements for an LLM to extract hypergraphs are high, and hypergraphs are hard to translate into LTL. In contrast, the triplet schema already contains sufficient information to accurately describe the event sequences.
>
> **Q5,6&W1,3**. Thanks for noting missing citations of prior works with similar reward design. We will include them in revision. We further clarify our advantages in automation and reward design. Compared to [2], our reward design is already rich. Leveraging LLM semantic understanding, our reward rules achieve a diverse reward mixture through multi-category LTLs. For instance, self-loops are handled via the sudden repeat category in negative LTLs.
>
> Regarding [1] (ARM-FM), its RM construction relies heavily on the environment APIs, requiring all explicit object and state knowledge. It is tightly coupled and requires heavy customization per environment; thus offers no higher automation than GLARE while maintaining its own APs for RM construction. [1] also explicitly acknowledges human intervention across tasks.
>
> We also conducted experiments on the WebShop. Adapting required only modifying AP extractor few-shot examples and relaxing the generation constraints for categories. The LTL Translator and Critic required no change, because their manually defined categories are generic, focusing on scenario-independent logical relations, enabling cross-task adaptation.
>
> Furthermore, by prompting LLMs to construct RMs from agent trajectories, we can dynamically generate richer RMs covering edge cases—unlike [1], which derives RMs from task descriptions.
>
> |Method|Env dependency|Handcraft|RM source|Reward type|
> |-|-|-|-|-|
> |ARM-FM| Full env APIs|Manual RM fixes|Task-based|Sub-goal only|
> |Adapt Rew|Pre-parsed states|Hand-crafted LTL rules|Hard-coded formulas|Hybrid|
> |GLARE|None|Custom few-shot|Trajectory-based|Hybrid|
>
> **Q7**. Certain traditional RL works [3] argue that using constraints as purely negative penalties is problematic. Furthermore, discussions within the LLM domain regarding KL/length penalties are notoriously difficult to control. They easily induce policy collapse, driving the model to execute overly conservative (do-nothing) behaviors. In GLARE, ensuring positive rewards exceed penalties balances both. We have consistently achieved superior performance in ALFWorld and WebShop under this design.
>
> [1]ARM-FM: Automated Reward Machines via Foundation Models for Compositional Reinforcement Learning, Castanyer, Roger Creus, et al.
>
> [2]Adaptive Reward Design for Reinforcement Learning, Minjae Kwon, Ingy ElSayed-Aly, Lu Feng
>
> [3]Penalizing side effects using stepwise relative reachability, Krakovna, Victoria, et al.

---

> > ### Author Rebuttal · Reviewer_z6zx · 2026-04-02
> >
> > Thank you for the authors to address my concerns. Except for the two questions, I state below, everything else is well addressed, and I am convinced.
> >
> > **Q1. Clarification on the claimed “richer” reward design.**
> > I disagree with the proposed reward design is richer than prior automaton-based reward shaping, even though I agree these reward design can be dense as you explained well on **Q1&W2** answer. However, At the notation level, the closest correspondence is between the prior work [2] 's automaton progress reward $\rho_{\varphi}\left(q, q^{\prime}\right)$ and the term $\lambda_{\text {pos }} \mathbf{1}\left(q_{t+1} \in P\left(q_t\right)\right)$ in this paper, just difference by constant. Both reward transitions that advance the automaton toward acceptance.
> >
> > Moreover, prior work's hybrid reward design seems to be more dense due to the self-loop penalty as below;
> > $$
> > R_h\left(\langle s, q\rangle, a,\left\langle s^{\prime}, q^{\prime}\right\rangle\right)= \begin{cases}
> > -\eta d_{\varphi}(q), & q=q^{\prime} \\\
> > (1-\eta) \rho_{\varphi}\left(q, q^{\prime}\right), & q \neq q^{\prime}
> > \end{cases}
> > $$
> >
> > In your formulation, the logic-guided reward is
> >
> > $$
> > r_t^{L T L}=\lambda_{\text {pos }} \mathbf{1}\left(q_{t+1} \in P\left(q_t\right)\right)+\lambda_{\text {trend }} \mathbf{1}\left(t \in T_{\text {trend }}\right)+\lambda_{\text {neg }} \mathbf{1}\left(\exists j: \mathcal{A}_j^{-} \text {rejects at } t\right) \text {, }
> > $$
> >
> > with
> > $R_t^{\text {total }}=r_t^{e n v}+\beta r_t^{L T L}+r_t^{f m t} .$
> > Also, I found that $r_t^{fmt}$ is not clearly described in the main page, even though I assume this is some reward for formatting.
> >
> > $\lambda_{\text {pos }} \mathbf{1}\left(q_{t+1} \in P\left(q_t\right)\right)$ seems to play the same role as the automaton progress reward, while the negative term appears analogous to a penalty for undesirable non-progress / violating behavior. Therefore, it is unclear in what precise sense the reward is "richer," as opposed to being another instance of automaton-based progress-plus-penalty shaping. Could the authors clarify more precisely what aspect is richer here: the scalar reward form itself, the logical specification used to generate reward events, or something else?
> >
> > I think this paper's core contribution is the system design to work with symbolic logics with RL and LLM agents, which I think it's a good contribution. And the proposed logic reward design is well studied in automaton fields, so claiming this reward design is novel makes me feel authors framing their contributions in a wrong way, in my opinion.
> >
> > **Q2. Why are multiple negative LTL formulas needed?**
> > The paper constructs one positive LTL formula together with multiple negative LTL formulas (e.g., 0-10 NegLTLs per task). However, under the current reward form, the negative contribution seems to depend only on whether any negative monitor rejects: $\lambda_{\text {neg }} \mathbf{1}\left(\exists j: \mathcal{A}_j^{-} \text {rejects at } t\right) .$
> >
> >
> > If so, it seems that multiple simple negative constraints such as
> >
> > $$
> > G(\neg A), \quad G(\neg B), \quad G(\neg C)
> > $$
> >
> > could be merged into a single safety formula
> >
> > $$
> > G \neg(A \vee B \vee C),
> > $$
> >
> > which would allow the method to track a single negative automaton state instead of multiple separate monitors. Because based on the reward form, one of violation happens, negative reward. This can be tracked by one monitor not several separate ones.
> >
> > Could the authors explain why multiple Neg-LTL monitors are actually needed, rather than using one combined negative LTL?
> >
> > I will postpone my re-evaluation for the scoring until the end. Again, I think the paper's core contribution is the system design, not these reward design, which I think valuable.
> >
> > ---
> > [2]Adaptive Reward Design for Reinforcement Learning, Minjae Kwon, Ingy ElSayed-Aly, Lu Feng

---

> > > ### Author Response · Authors · 2026-04-08
> > >
> > > We sincerely thank the reviewer for the further discussion; your follow-up comments and questions help us better clarify the statement of the contributions of our work. Indeed, our core contribution is the system design to work with symbolic logics with RL and LLM agents. We acknowledge, as the reviewer pointed out, that our reward shaping form (automaton-based progress-plus-penalty) follows the paradigm established in​ prior works. We will explicitly cite them to accurately articulate our core contributions.
> > >
> > > **Richer reward event**: However, we would like to point out that GLARE is richer in the logical specification used to generate reward events. Specifically, GLARE expands the semantic coverage of the positive progress set $P(q_t)$ and the negative monitors $A_j^-$ used in the reward terms $1\left(q_{t+1} \in P\left(q_t\right)\right)$ and $1\left(\exists j : A_j^- \text{rejects at } t\right)$.
> > >
> > >  1) We leverage the semantic understanding capabilities of LLMs to dynamically generate various forms of reward events ($P(q_t)$ and $A_j^-$), rather than being limited to static rewards for specific tasks. This provides our framework with cross-task adaptability, enabling the generation of targeted reward events without the need for manual, task-specific reward engineering.
> > >
> > >  2) As shown in the examples in Appendix F, Figure 14, we demonstrate multiple forms of Neg-LTL (e.g., `sudden_repeat`, `lack_premise`). Our Neg-LTL offers a richer form of expression ( which enrichs $A_j^-$ ) than purely penalizing a lack of progress (self-loop). This enriched expressiveness empowers our system with precise, fine-grained credit assignment. Rather than receiving a generic penalty for lacking progress, the agent receives actionable supervision that pinpoints the exact logical fallacy at a specific step.
> > >
> > > **What is $r^{fmt}_t$**: $r^{fmt}_t$ represents the format penalty for the LLM's output. For example, we require the LLM to output in the `<think>...</think><action>...</action>` format to facilitate the extraction of specific actions. If the model does not strictly adhere to this output format, a penalty is applied. This reward term is primarily effective in the early stages of training, as it enables the model to reach very high formatting accuracy in just a few steps.
> > >
> > > **Multiple negative LTL**: Indeed, multiple formulas can be merged, and a merged single Neg-LTL formula and multiple independent Neg-LTL formulas are essentially identical at runtime when constructing the automaton. We chose the multiple-formula format because modular management offers engineering advantages: **1) Stability**: Since the critic explicitly enumerates bad behaviors one by one, the subsequent LTL translator maintains an "independent behavior to independent formula" mapping in format, reducing the error rate of the LLM when generating complex nested logic across different formulas. It also achieves local fault tolerance: if an independent LTL fails to compile, it is simply discarded without causing a global failure. **2) Intuitive Interpretability**: Independent formulas make it easy to track the specific reasons for penalties, providing higher interpretability. Furthermore, since Neg-LTL usually involves temporal logic rather than simple prohibitions, independent formulas avoid the lengthy and unreadable formulas that result from forcibly splicing temporal operators.

---

### Official Review · Reviewer_DYXA · 2026-03-13

**Soundness:** 3
**Presentation:** 3
**Significance:** 3
**Originality:** 3
**Overall Recommendation:** 5
**Confidence:** 3

**Summary:**

This paper introduces GLARE, a new neuro-symbolic reward shaping framework for RL and LLM agents. The challenge addressed is sparse and unstable rewards in long-horizon tasks. The key idea is to separate semantic interpretation from credit assignment by translating trajectories into LTL formulas. The logic formulas are then used to generate new reward signals to help mitigate the sparsity of the original reward. Empirical evaluations are conducted on the ALFWorld benchmark and show improvements using GLARE.

**Compliance With Llm Reviewing Policy:**

Affirmed.

**Final Justification:**

The authors convinced me of the quality of their work.

**Key Questions For Authors:**

1. How sensitive is GLARE to errors in event extraction?
2. What is the computational overhead of the extraction and automata construction pipeline?
3. Did you test the scalability of your automata representations with respect to task complexity?
4. Would you argue that GLARE is specific to this type of task, or could it be transferred to other environments? What challenges might arise in such cases?

**Limitations:**

Yes, the limitations are addressed.

**Strengths And Weaknesses:**

## Strengths
+ The idea of translating trajectories into logic formulas is highly interesting and opens doors to many new research directions, as compiling automata to provide structured rewards is definitely an appealing way to combine semantics with reinforcement learning and large language models.
+ Since sparse rewards are a well-known problem in reinforcement learning, the proposed approach represents a valuable and direct attempt to address this challenge.
+ Symbolic automata are not very computationally expensive at runtime, which keeps the overhead minimal while also providing an easy way to verify and interpret the reward function.

## Weaknesses
- The framework relies heavily on the accurate extraction of semantics to build these automata, but the paper does not sufficiently address the limitations and potential failures of this crucial step.
- Generating the Linear Temporal Logic (LTL) formulas is a required step for this work, and while the paper suggests an automatic generation method that works for ALFWorld, it is not fully clear how this automated process would translate to different evaluation environments.
- Because the related work section focuses very heavily on RL and LLM approaches, it would significantly strengthen the paper to include a theoretical comparison with other structured reward shaping methods or prior neuro-symbolic reinforcement learning techniques.
- The generality of the proposed reward framework is difficult to fully assess because the evaluation focuses exclusively on a single, widely used environment that is specific to embodied language tasks.

### Further Limitations
+ Because the approach relies on accurate semantic event extraction from trajectories, any errors or ambiguities in this step may lead to incorrect symbolic representations that could propagate negatively into the reward shaping mechanism.
+ The framework depends on generating appropriate LTL specifications for the task, and while the paper proposes automated generation, the scalability and reliability of this process across diverse environments and complex task structures remain unclear.
+ The automata-based reward shaping inherently encodes specific task structures, which may introduce biases toward predefined trajectories and potentially limit exploration or prevent the agent from discovering alternative valid strategies.
+ The evaluation is conducted primarily on ALFWorld, making it difficult to confidently assess how well the framework generalizes to entirely different domains.

### Suggested Improvements
1. The authors should discuss the method's limitations and its broader applicability much more clearly within the text.
2. If a figure is placed prominently on the first page, it must be properly referenced in the text and incorporated meaningfully into the paper's core argumentation.
3. The paper should explicitly address the topic of handling imperfect symbolic specifications in order to strengthen the real-world applicability of the approach.
4. Providing a clearer overview of hyperparameters, computational costs, and additional experimental details is necessary to make reproducibility easier and to substantiate the claims of a fair comparison.

---

> ### Author Rebuttal · Authors · 2026-03-31
>
> We thank the reviewer's constructive feedback and address your concerns below.
>
> **Q1&W1&L1** Semantic extraction operates at two distinct levels: atomic proposition (AP) extraction and milestone/bad behavior extraction. By enforcing structured triplet extraction via the LLM with few-shot prompting, and by sharing the extraction results for identical states across different trajectories, we ensure that the reliability of using LLMs to extract triplets and process them into APs is extremely high, which is evaluated in Sec. 6.3. Conversely, LLM-as-a-judge requires identifying score-contributing events/attributes; this process is inevitably affected by the foundation model's potential errors, a systematic bottleneck shared by all such paradigms. Similarly, the extraction of milestones/bad behaviors depends on the intrinsic capabilities of the judge model. GLARE is sensitive to errors in event extraction, but our goal is not to enhance the judge model's analytical and comprehension capabilities, rather to unlock the model's potential by decoupling semantic analysis from the final scoring.
>
> **Q2**
> AP Extractor and LTL Translator use small models with negligible cost compared to the central critic, which is approximately 25000 and 20000 tokens per step. While these absolute token counts are not significantly lower than that of the central critic, these two components can be effectively powered by much more lightweight models in practical applications. Consequently, their actual computational and financial overhead remains extremely low. In the main text, we only reported the token consumption of the critic—which is typically the most expensive—to demonstrate our cost advantage regarding the central critic.
>
> **Q3&L2** The ALFWorld benchmark inherently comprises tasks of varying complexity, ranging from simple single-object retrieval (PICK) to intricate multi-object interactions (PICK2). To demonstrate scalability, we statistically analyze the properties of the generated LTL formulas across these different task categories. In response to increased task complexity, GLARE adaptively generates more LTL formulas to implement richer constraints and milestones. We conducted a statistical analysis of the LTL formulas for both PICK and PICK2 tasks: in PICK2, the number of milestone stages described by positive LTL formulas is 2.4 times that of PICK, while the number of negative LTL formulas increases by 1.6 times. Crucially, we observed no significant degradation in formula compilation success rates despite this increased complexity. This evidence demonstrates both the scalability and reliability of GLARE across diverse levels of task difficulty.
>
> **Q4&W2,4&L2,4** We have supplemented the experiments on WebShop (a web navigation task, different from the embodied tasks in ALFWorld). Generating the LTL formulas involves three components. Among them, LTL Translator and Critic (used for identifying behaviour and generate LTL formulae) required zero modifications: the milestone extraction logic relies entirely on the critic model's autonomous judgment. Similarly, the predefined error categories are scenario-independent, focusing purely on the logical relationships of events, and the subsequent LTL translation logic is also domain-agnostic. We only replaced the few-shot examples in the AP Extractor, retaining the LLM-based triplet AP extraction but relaxing the category constraint to allow free generation. This demonstrates the extensibility of GLARE across different tasks. Experimental results is above.
>
> |Method|Score|Succ|
> |-|-|-|
> |GRPO|75.8|56.8|
> |LLM Judge|78.8|61.2|
> |GLARE|80.2|63.4|
>
> **Comparison with neuro-symbolic RL & limit exploration(W3&L3)**1) Traditional neuro-symbolic RL[1,2] typically relies on manually crafted rules and is task-specific specification (whereas the task types in ALFWorld and WebShop are highly diverse). In contrast, we introduce the semantic understanding capabilities of LLMs to generate targeted LTL formulas based on actual trajectory data, innovatively realizing an "LLM->LTL->LLM" training paradigm. We will add a detailed comparative discussion with these methods in our revised version. 2) Furthermore, based on this paradigm, GLARE's LTL library is "dynamically generated". While static automaton-based rewards indeed encode specific task structures, our dynamic formulation avoids introducing biases toward predefined trajectories. If the agent discovers an unconventional but valid new path, the Central Critic will recognize this novel strategy an generate a new Milestone LTL for it, thereby not limiting exploration. We will incorporate this discussion in the revised version.
>
> **Presentation**We will cite Figure 1 in the introduction for the core argumentation and add more detailed experimental setups in the appendix.
>
> [1]Reward machines: Exploiting reward function structure in reinforcement learning. Icarte, et al
>
> [2]Neurosymbolic reinforcement learning and planning: A survey. Acharya, et al

---

> > ### Author Rebuttal · Reviewer_DYXA · 2026-04-04
> >
> > I thank the authors for their answers, I am maintaining my score.

---

### Official Review · Reviewer_y8Hc · 2026-03-13

**Soundness:** 2
**Presentation:** 2
**Significance:** 3
**Originality:** 3
**Overall Recommendation:** 4
**Confidence:** 4

**Summary:**

This paper presents GLARE, a neuro-symbolic reward framework to train LLM agents with fine-grained rewards. Rather than relying solely on sparse outcome rewards or LLM judges, the method maps trajectories to sequences of discrete events, which are interpreted as logical propositions. Together with the trajectory outcomes, these propositions are then translated into LTL formulae by an LLM judge model. These formulae capture common high-level milestones and negative behaviour that leads to success or failures. The method then employs standard LTL-to-automata translations to obtain DFA representations of the formulae, which are used to construct a step-level shaped reward signal. Finally, these rewards are optimised with an adapted, step-level GRPO scheme.

**Compliance With Llm Reviewing Policy:**

Affirmed.

**Final Justification:**

The authors engaged well in the rebuttal, which addressed my main concerns. The new results on WebShop make the paper significantly stronger and demonstrate the applicability of the method, and the authors convinced me that their method uses adequately expressive LTL formulae.

There are some inconsistencies in the paper regarding the difference between step-level advantage (grouping by timestep within the group) and GiGPO-style step-relative advantage (anchor-state grouping). I encourage the authors to carefully revise the explanation of these aspects in the final version of the paper. Furthermore, there is no experiment that clearly ablates the effect of the grouping mechanism used in GLARE: as far as I understand, GLARE computes advantage means over the entire group across timesteps, whereas the baselines group by timestep.

Another concern of mine was the comparison with GiGPO (with anchor-state grouping). The authors provided this comparison in the rebuttal, which showed that GiGPO outperforms GLARE when anchor-state grouping is applicable. At the same time, the authors presented additional results in a modified AlfWorld experiment where anchor-state grouping is not applicable, demonstrating that GLARE remains effective whereas GiGPO performs significantly worse.

Overall, I agree with the authors that the main contribution of the work is the overall framework, which seems reasonably novel and (with the new results) convincing. I encourage the authors to include clean ablations of the grouping component and the comparison with GiGPO in the revised paper.

**Key Questions For Authors:**

See weaknesses.

**Limitations:**

yes

**Strengths And Weaknesses:**

**Strengths**

- The proposed approach cleanly separates semantic analysis of trajectories from credit assignment. A judge model fist analyses trajectories to find common positive and negative patterns and these are then automatically rewarded or penalised at the corresponding timesteps based on the DFA constructed from the LTL formula.
- Separating trajectory rollout from their analysis and scoring is conceptually clean and makes sense.
- The experimental results provide several interesting insights, such as token efficiency and inconsistencies in standard LLM judge-based rewards.

**Weaknesses**

The main weakness of the paper is the limited experimental evaluation. Results are only provided on ALFWorld, which is an interesting benchmark, but nevertheless relatively simple in terms of observations and tasks. It is unclear how well the method would translate to more complex, real-world benchmarks such as web browsing, where extracting atomic propositions and identifying behaviour via LTL formulae is significantly more challenging. This is especially true since some of the prompts (e.g. proposition extraction) are specifically engineered towards the ALFWorld benchmark.

Furthermore, the experimental evaluation omits GiGPO (Feng et al., 2025) as a crucial baseline. Despite using a very similar step-level credit assignment mechanism (without citing the GiGPO in the relevant section), the method is mainly compared to variants of standard GRPO and PPO. This makes it difficult to evaluate how much of the reported improvements are due to the LTL-based rewards, and how much can be attributed to the step-level advantages that already appeared in previous work, and demonstrated significant performance improvements.

The advantage computation in Eq. (8) disregards possible (and likely) state divergences between different trajectories. By grouping trajectories based solely on the timestep $t$, but not by the actual MDP state $s$ they are in at that timestep, the advantage computation is inherently biased. This should be explicitly acknowledged in the paper. Have the authors explored any state-based grouping mechanisms to address this bias?

While employing LTL as a mechanism for step-based rewards is conceptually nice, the examples shown in the paper beg the question if the full power of LTL is really needed, or if simple sequences of propositions with safety constraints are already sufficient. Can the authors provide notable examples of LTL formulae constructed by their procedure? Do these capture interesting behaviour, or mainly sequential reaching of milestones?

Furthermore, while the analysis in Section 5.4 shows that the method is generally more token-efficient than the baselines, the proposed approach never reuses LTL formulae constructed from previous trajectories. This seems wasteful, and the contribution of the paper would be much stronger if the method could successfully reuse LTL formulae, instead of having to re-analyse and compute new formulae for every rollout stage.

There are also several presentation issues:
* The background section on LTL should provide references for unfamiliar readers. Some statements such as "any LTL formula [..] can be translated into a deterministic Finite State Automaton" are incorrect; it is well-known that (full) LTL requires nondeterministic automata.
* It seems like formulae are always evaluated over finite traces; in this case, the approach would be theoretically cleaner if based on the finite-trace variant LTL$_f$ (which also sidesteps the issue of nondeterminism).
* Section 4.2 reads a bit generic and does not include enough detail to understand how exactly LTL formulae are constructed from trajectories, which is crucial. Some components (e.g. "counterfactual reasoning mechanism") are not explained at all.
* Section 5.3 mentions significant degradation in baseline performance with continued training, yet no numbers or plots showing this phenomenon are reported.
* There are some minor spelling and grammar mistakes throughout.

---

> ### Author Rebuttal · Authors · 2026-03-31
>
> Thank you for the insightful suggestions. We address and clarify your specific concerns as follows.
>
> **Additional Experiment & Comparsion(W1,2)**: 1) We migrated GLARE to WebShop—a web browsing task fundamentally distinct from ALFWorld—to validate the framework's generalizability. 2) The core LTL Translator and Critic (used for identifying behaviour and generate LTL formulae) required zero modifications: the milestone extraction logic relies entirely on the critic model's autonomous judgment. Similarly, the predefined error categories are scenario-independent, focusing purely on the logical relationships of events, and the subsequent LTL translation logic is also domain-agnostic. Thus, it naturally adapts to object-centric scenarios, which is not specifically engineered towards specific benchmark. We only replaced the few-shot examples in the AP Extractor, retaining the LLM-based triplet AP extraction but relaxing the category constraint to allow free generation (typically product attributes like size and color, which are definitive in WebShop).
>
> 3）Furthermore, we fully agree with incorporating GiGPO into our discussion as a crucial baseline. However, we must point out the methodological differences between GiGPO and GLARE. While GiGPO achieves impressive empirical success, its fine-grained grouping heavily relies on the exact matching of underlying environment states. Conversely, GLARE's neuro-symbolic abstraction (via APs) bypasses this strict matching requirement, offering robust observation invariance alongside explicit white-box interpretability. Regarding the step-level advantages, we emphasize that all our training—including the GRPO and LLM-as-a-judge baselines—consistently utilized the step-level advantage proposed by GiGPO. This ensures fair comparison: GLARE's performance gains are entirely attributable to our LTL-based reward design. We will explicitly cite GiGPO and clarify this in the revised version.
>
> The WebShop results and methodological comparisons are summarized below:
>
> |Method|Exact state match rely|Interpretability|Step-level adv|Score | Succ.|
> |-|-|-|-|-|-|
> |GRPO|x|x|√|75.8|56.8|
> |GiGPO|√|x|√|83.1|65.0|
> |LLM Judge|x|√|√|78.8|61.2|
> |GLARE|x|√|√|80.2|63.4|
>
> **Writing Mistake(W3)**: Upon inspection, we found that there is indeed an issue with the formula in Eq. (8). Neither our original intention nor our actual implementation involves alignment by timestep; the advantage is calculated for a single step relative to all steps within the group. We will remove this erroneous statement in the revised manuscript. Our experimental results remain unaffected, which can be verified by checking the loss computation section (core_trpo.py->compute_trpo_loss) in the code from our initially submitted supplementary materials.
>
> **Necessity of LTL(W4)**:
> Simple proposition sequences(A->B->C...)capture only straightforward sequential dependencies. LTL's primary advantage lies in its ability to handle temporal branching and nested logical relationships, which are difficult to capture with simple "if-then" sequences. For instance, in the specific formula:
> `F((obs_self_loc_bed1 & (obs_laptop1_loc_bed1 | obs_laptop2_loc_bed1)) & F((¬obs_self_state_notcarrying & obs_self_loc_desk1 & obs_laptop2_loc_desk1)))`
> These milestones incorporate branching (either laptop 1 or 2) and joint constraints. In the third phase, the formula requires the laptop, the agent, and the target location to coincide while the agent is not carrying any other object. This logical nesting ensures that the milestone is only triggered by legitimate task progress, effectively preventing the agent from "hacking" rewards via unintended or partial state transitions.
>
> Further, GLARE employs combinations of LTL operators to express compositional temporal logic relationships beyond simple forbid. Appendix F (Fig.14) shows temporal dependencies like G(act_A -> X(!act_A)) to prevent immediate repetitive actions and G(!obs_A -> !act_B) to enforce preconditional forbid.
>
> **LTL Reuse(W5)**: The idea of reusing LTL formulas is highly valuable. However, within our current framework, continuously processing changing trajectory groups remains indispensable to reveal new errors and milestones; reusing old LTL formulas still cannot circumvent the need to process the trajectory data again. We will explore LTL formula reuse mechanisms to further enhance token efficency in future work.
>
> **Presentation Issues**:
> 1) We fully adopt your suggestion to improve the LTL-related citations and descriptions, and we will rigorously correct the terminology to $LTL_f$ (LTL on finite traces), which perfectly aligns with our underlying code logic.
> 2) We will provide a practical example in the appendix to demonstrate how LTL formulas are constructed from trajectories, clearly illustrating the function of each module.
> 3) We will add specific numbers (76.4 -> 59.7 from 100 steps to 150 steps) in the section mentioning 'degradation' and correct the spelling and grammar mistakes.

---

> > ### Author Rebuttal · Reviewer_y8Hc · 2026-04-03
> >
> > I thank the authors for their thorough response, which has addressed some of my concerns. The new results on WebShop seem promising, and evaluating on more than one environment makes the paper significantly stronger, and the shown formula nicely demonstrates different aspects of LTL being used.
> >
> > **GiGPO baseline and reward design**
> >
> > Despite the clarification by the authors, this aspect still is not completely clear to me, and I think it's important to clarify.
> >
> > > Regarding the step-level advantages, we emphasize that all our training—including the GRPO and LLM-as-a-judge baselines—consistently utilized the step-level advantage proposed by GiGPO.
> >
> > Can you clarify which precise step-level advantage was used for GRPO and LLM-as-a-judge? If you use the advantage proposed by GiGPO, why is there a difference in the reported results between GRPO and GiGPO on WebShop? Please clarify the exact instantiation of GRPO used as the baseline.
> >
> > > Upon inspection, we found that there is indeed an issue with the formula in Eq. (8). Neither our original intention nor our actual implementation involves alignment by timestep; the advantage is calculated for a single step relative to all steps within the group.
> >
> > I appreciate the clarification. However, if I understand the your comment correctly, my original criticism still stands (and is in fact even amplified): by computing the advantage of an action in one step relative to all other steps in the group, the method does not account for state divergences over the course of the different rollouts. Hence, the computed advantage estimate is likely not a good estimate of the real advantage A(s,a), but instead just the reward modified with a constant batch-level baseline. Can the authors clarify the motivation for this choice of advantage computation?
> >
> > Furthermore, the advantage only accounts for the immediate reward, rather than episode-level returns, which biases the policy gradient towards myopic high-reward actions rather than long-term successful trajectories. Both of the above are important theoretical and practical concerns.
> >
> > > WebShop results
> >
> > The new WebShop results show GiGPO performing significantly better than GLARE. I appreciate that, in contrast to GLARE, GiGPO relies on exact state matching. However, this is easily possible in the considered environments (both AlfWorld and WebShop). The experimental results would be much stronger if the authors demonstrated the advantages of their method in environments where GiGPO is not readily applicable, or fails due to a large number of reachable states. Additionally, can the authors also provide GiGPO results on AlfWorld?
> >
> > I also noticed the authors did not include the results of GiGPO outperforming GLARE in the responses to the other reviewers, which seems a bit misleading.
> >
> > Overall, the step-level advantage remains a crucial component to properly ablate, especially given the confusion around how it was applied to the baselines. To isolate the contribution of the LTL-reward from the advantage computation, could the authors provide (i) a clean ablation of GLARE using standard episode-level GRPO advantage (no step-level advantage), and (ii) a clear breakdown of the exact algorithmic differences between the GiGPO and GRPO lines reported in the new WebShop results, assuming both used GiGPO's advantage computation as claimed?

---

> > > ### Author Response · Authors · 2026-04-08
> > >
> > > We sincerely thank the reviewer for their constructive suggestions. Here, we address the left concerns below.
> > >
> > > **What is step-level advantage**: When we refer to `step-level advantage`, we are specifically describing the granularity of the advantage computation. This is inherited from `verl-agent` (training framework provided by the authors of GiGPO). It calculates the advantage among steps within the same group (formed during rollout), as opposed to conventional GRPO which directly computes advantages at the episode level. In `verl-agent`, GRPO can be trained with this `step-level advantage`, which can enhance the training stability of multi-turn dialogues (as the REINFORCE++[1] suggests that normalizing over a larger batch yields greater stability, even across different states).
> > >
> > > However, the core contribution of GiGPO paper lies in utilizing identical steps across a batch for anchor state grouping (rather than grouping during rollout) to compute an additional `Step Relative Advantage`. This is its primary distinction from GRPO with `step-level advantage`. In our experiments, GLARE, the LLM-as-a-judge baseline, and the base GRPO did not introduce this `Step Relative Advantage`, but with `step-level advantage`. This ensures that our experimental comparisons are fair. By utilizing this design, our performance gains stem solely from the dense reward signals rather than algorithmic variations.
> > >
> > > We acknowledge that our previous presentation in paper and rebuttal statement may indeed cause misunderstanding. Thanks for pointing this out. We will explicitly clarify this conceptual distinction in the revised manuscript.
> > >
> > > **Why use step-level advantage**: We agree with the reviewer that in multi-turn, critic-free scenarios, it is infeasible to accurately estimate the state value $V(s_t)$ (which requires strict state-based grouping or independent critic to estimate) for each step to compute the exact advantage $A(s,a)$. However, directly accumulating step-level rewards into an episode-level reward obscures intra-episode credit assignment, which loses the fine granularity of step-level rewards.
> > >
> > > Given this observation, we draw inspiration from [1] to construct an effective surrogate credit signal for policy optimization to leverage the fine granularity of step-level rewards. We treat each step-level reward as locally meaningful feedback to characterize the action's positive or negative contribution to task progress (e.g., whether it violates a constraint or achieves a key milestone). Subsequently, we apply group-level normalization: using the in-group step mean as a baseline to reflect the overall performance of the current policy, while stabilizing gradients through standardization.
> > >
> > > Furthermore, in our reward design, each step receives a combined reward (Eq.7): $R_t^{\text{total}} = r_t^{\text{env}} + \beta \cdot r_t^{\text{LTL}} + r_t^{\text{fmt}}$. The $r_t^{\text{env}}$ dominates this combination, ensuring that the overall optimization direction remains strictly aligned with final task success, preventing the agent from falling into myopic optimization driven solely by local signals.
> > >
> > > Ultimately, our core contribution lies in the system architecture—specifically, introducing LTL-based reward generation for the training of LLMs. Other advanced algorithms designed to better utilize step-level rewards can naturally be integrated with our framework as orthogonal contributions.
> > >
> > > **Additional Experiment**: Following the reviewer's insightful suggestion, we compare GLARE with GiGPO in a non-exact state matching setting. We prompted an LLM to rewrite original ALFWorld observations by integrating them with randomly sampled distractor combinations (drawn from a pool of 1,200: 10 descriptions $\times$ 15 items (that do not exist in ALFWorld to avoid interfering with the extraction of actual object states) $\times$ 8 locations), while strictly preserving core names and relations. This modification only changes whether exact state matching is available, while leaving other aspects unchanged. In this setting, GiGPO’s state-anchoring fails, whereas GLARE remains robust. We present the results below, and report GiGPO's performance on the original ALFWorld (Exact state matching). In the revision, we will explicitly acknowledge this performance gap, while highlighting GLARE's advantage of being state-anchor-free.
> > >
> > > |Method|Size|Exact state matching in ALFWorld|Succ|
> > > |-|-|-|-|
> > > |GiGPO|1.5B|√|86.7|
> > > |GiGPO|7B|√|90.8|
> > > |GiGPO|1.5B|x|71.4|
> > > |GLARE|1.5B|x|83.3|
> > >
> > > [1] Reinforce++: A simple and efficient approach for aligning large language models. Hu, Jian

---

### Decision · Program_Chairs · 2026-04-30

**Decision:**

Accept (regular)

**Comment:**

This paper proposes a novel algorithm for devising shaped rewards for LLM reinforcement learning by converting trajectories into sequences of discrete events, and then tracking task progress using deterministic automata. The reviewers generally found the approach to be compelling. There were several concerns about the scope of the experiments, especially limited datasets and missing baselines; these concerns were largely addressed during the rebuttal period. There were some remaining concerns about the generalizability of the approach, which was partially addressed by the new WebShop benchmark though additional benchmarks would further strength these claims.